

# Carbon cycle and climate feedback under CO₂ and non-CO₂ overshoot pathways

Irina Melnikova[1,2], Philippe Ciais[2], Katsumasa Tanaka[1,2], Hideo Shiogama[1], Kaoru Tachiiri[1,3], Tokuta Yokohata[1] and Olivier Boucher[4]

[1]Earth System Division, National Institute for Environmental Studies (NIES), Tsukuba, 305-8506, Japan,
[2]Laboratoire des Sciences du Climat et de l'Environnement (LSCE), IPSL, CEA/CNRS/UVSQ, Université Paris-Saclay, Gif-sur-Yvette, 91191, France
[3]Research Institute for Global Change, Japan Agency for Marine-Earth Science and Technology, Yokohama, 236-0001, Japan
[4]Institut Pierre-Simon Laplace, Sorbonne Université / CNRS, Paris, 75005, France

*Correspondence to*: Irina Melnikova (melnikova.irina@nies.go.jp)

**Abstract.** Reducing emissions of non-CO₂ greenhouse gases complements CO₂ mitigation in limiting global warming. However, estimating carbon-climate feedback for these gases remains fraught with uncertainties, especially under overshoot scenarios. This study investigates how CO₂ and non-CO₂ gases with nearly equal effective radiative forcing magnitudes impact the climate and carbon cycle using the Earth System Model IPSL-CM6A-LR. We first present a method to recalibrate methane and nitrous oxide concentrations to align with published radiative forcings, ensuring accurate model performance. Next, we carry out a series of idealised ramp-up and ramp-down concentration-driven experiments and show that while the impacts of increasing and decreasing CO₂ and non-CO₂ gases on the surface climate are nearly equivalent (when their radiative forcing magnitudes are set to be the same), regional differences emerge. We further explore the carbon cycle feedback and demonstrate that they differ under CO₂ and non-CO₂ forcing. CO₂ forcing primarily affects temperature-driven feedback, whereas non-CO₂ gases influence both temperature and carbon-concentration feedback. We introduce a novel framework to separate the carbon-climate feedback into temperature and cross terms, revealing that these components are comparable in magnitude for the global ocean. This highlights the importance of considering both components in Earth System modelling and climate change mitigation strategies.

**Plain language summary.** Mitigation of non-CO₂ greenhouse gases complements CO₂ mitigation in limiting global warming. We investigate the effects of reducing emissions of non-CO₂ greenhouse gases and compare them with CO₂ mitigation using an Earth System Model. Both types of gases contribute to global warming, but their impacts on climate vary regionally. Furthermore, we highlight that the carbon cycle feedback differ between CO₂ and non-CO₂ gases, with the presence or absence of CO₂ change in the atmosphere influencing their effects. The study underscores the need to consider interactions between CO₂ and non-CO₂ impacts on the carbon cycle in climate models and emission reduction strategies.



# 1 Introduction

Increases in the atmospheric concentrations of carbon dioxide ($CO_2$), methane ($CH_4$), and nitrous oxide ($N_2O$), predominantly cause human-induced climate change since the preindustrial period. They contributed nearly 63%, 11% and 6%, respectively, to the total effective radiative forcing (ERF) of the 1960–2019 period (Canadell et al., 2021). Anthropogenic $CO_2$ emissions

are dominated by the combustion of fossil fuels (FF) and land-use change (LUC), $CH_4$ emissions by FF and the agricultural sector, and $N_2O$ emissions by the use of nitrogen fertiliser and manure. $CH_4$ and $N_2O$ have atmospheric lifetimes of $11.8 \pm 1.8$ years and $109 \pm 10$ years and 100-year global warming potentials over (GWP100) of 27.9 and 273, respectively (Forster et al., 2021; Myhre et al., 2013). Mitigation of non-$CO_2$ GHGs is an essential strategy to limit global warming in the context of the Paris Agreement's temperature target (Abernethy et al., 2021; Jones et al., 2018; Rao and Riahi, 2006; de Richter et al., 2017;

Tanaka et al., 2021). The reduction of $CH_4$ emissions can lead to a rapid decrease in the radiative forcing and may limit the peak warming (Mengis and Matthews, 2020; Montzka et al., 2011). To facilitate the achievement of the Paris Agreement temperature target, the Global Methane Pledge was adopted to reduce anthropogenic $CH_4$ emissions by 30% over the 2020–2030 period (CCAC, 2021). Several existing studies have confirmed the technical and socioeconomic capacities to reduce the global methane emissions and benefits for reducing atmospheric pollution (Höglund-Isaksson et al., 2020; Jackson et al., 2021;

Malley et al., 2023; Nisbet et al., 2020). Atmospheric methane removal methods have also been discussed in some studies (Boucher and Folberth, 2010; Jackson et al., 2021; Mundra and Lockley, 2023) but they are still in their infancy.

Numerous studies have investigated the impacts of changes in the emissions / concentrations of $CO_2$ and non-$CO_2$ GHGs on the Earth System. The Precipitation Driver and Response Model Intercomparison Project (PDRMIP) focused on the role of different climate change drivers on the mean and extreme precipitation changes using a set of idealised perturbed experiments

(Myhre et al., 2017). Richardson et al. (2019) revealed spatial and temporal differences in the surface temperature response to different forcing components, in part due to the physiological $CO_2$ warming over the densely vegetated regions that is absent under non-$CO_2$ forcing. This physiological warming occurs because plants close their stomata under elevated $CO_2$, reducing transpiration and decreasing latent heat loss which causes a local surface warming. Nordling et al. (2021) further demonstrated that change in top of the atmosphere (TOA) long-wave clear-sky emissivity is the key driver for the temperature response

differences between GHGs forcings. Using Earth System model simulations, Tokarska et al. (2018) showed that non-$CO_2$ forcing reduces the remaining carbon budget due to its direct radiative effects on surface temperature, causing additional warming. Fu et al. (2020) showed that non-$CO_2$ GHG that have a shorter atmospheric lifetime than $CO_2$ may have long-term consequences on climate via their impacts on carbon cycle. Namely, warming from non-$CO_2$ GHG weakens land and ocean carbon sinks.

Previous studies have examined the impact of declining atmospheric $CO_2$ concentrations on the climate and carbon cycle (Boucher et al., 2012; Chimuka et al., 2023; Jones et al., 2016; Koven et al., 2023; Melnikova et al., 2021; Schwinger and Tjiputra, 2018). Melnikova et al. (2021) showed an amplification of the carbon-concentration ($\beta$) and carbon-climate ($\gamma$) feedback (the responses of the land and ocean carbon uptake to the changes in $CO_2$ concentration and climate change,





respectively) under decreasing $CO_2$ concentration and temperature. The effectiveness of non-$CO_2$ mitigation has been explored
and is an integral part of the integrated assessment models (Ou et al., 2021; Rao and Riahi, 2006; Tanaka et al., 2021). However,
few studies investigated the effects of declining non-$CO_2$ GHG concentrations on the climate and carbon cycle using Earth
System Models (ESMs). Abernethy et al. (2021) used an ESM to demonstrate the effectiveness of methane removal in reducing
global mean surface temperature, complementing negative $CO_2$ emissions. The purpose of this study is twofold:
to clarify whether the climate response to declining $CO_2$ and non-$CO_2$ differ globally and in different regions,
to investigate the carbon cycle feedback, especially the γ feedback under $CO_2$ and non-$CO_2$ GHG decrease, globally and
regionally, and its implication for climate change mitigation.
In this study, we conduct a set of idealised $CO_2$ and non-$CO_2$ ($CH_4$ and $N_2O$) concentration-driven ramp-up and ramp-down
experiments using the IPSL-CM6A-LR ESM. We then compare the global and spatial impacts of $CO_2$ and non-$CO_2$
concentration changes on climate and the carbon cycle, with a focus on carbon-climate feedback, under overshoot pathways.
Although the IPSL-CM6A-LR model, like many contemporary models, lacks interactive modules for the $CH_4$ and $N_2O$ cycles,
the use of the model is justified for several reasons. Current changes in $CH_4$ and $N_2O$ concentrations are primarily driven by
anthropogenic sources, suggesting that the absence of interactive modules of natural sink/source processes does not
significantly affect the representation of natural variability trends for the $CH_4$ and $N_2O$ concentration (Nakazawa, 2020;
Palazzo Corner et al., 2023; Zhu et al., 2013). Our understanding of the interactions of the emitted non-$CO_2$ GHGs with land
and ocean systems remains limited (Palazzo Corner et al., 2023; Yu et al., 2022). Furthermore, given the dominant influence
of climate factors, driven by the changes in radiative forcing, on the $CH_4$ and $N_2O$ fluxes, it is reasonable to expect that the
model is appropriate to address the objectives of this study.

## 2 Data and Methods

### 2.1 Recalibration of model's $CH_4$ and $N_2O$ concentrations

We use Version 6 of the Institut Pierre-Simon Laplace (IPSL) low-resolution ESM, IPSL-CM6A-LR, developed in the runup
of the sixth phase of the Coupled Model Intercomparison Project (CMIP) (Boucher et al., 2020). It comprises the LMDZ
atmospheric model Version 6A and the ORCHIDEE land surface model Version 2.0 with a 144×143 spatial resolution, and
the NEMO oceanic model Version 3 with a resolution of 1°.
Previous studies showed that IPSL-CM6A-LR can adequately estimate the ERF of $CO_2$ (Lurton et al., 2020). However, it
underestimates the $CH_4$ radiative forcing for the historical period due to known limitations in the parameterization of gaseous
optical properties in the Rapid Radiative Transfer Model (see Fig. 8 in Hogan and Matricardi (2020)). $CH_4$ also absorbs in the
shortwave spectrum, an effect that is not accounted for in the radiative transfer code used in IPSL-CM6A-LR and many other
climate models. The Intergovernmental Panel on Climate Change (IPCC) Sixth Assessment Report (AR6) (Forster et al., 2021)
estimated the $CH_4$ ERF to be 0.54 [0.43 to 0.65] W m$^{-2}$ for the 1750–2019 period. However, the estimated ERF of $CH_4$ at the
TOA in IPSL-CM6A-LR is only 0.27 Wm$^{-2}$ for the 1850–2014 period. Note that in the above estimates, the ERF is defined as



the difference in the net TOA flux between the perturbed experiments, in which model simulation setup includes fixed sea surface and ice temperatures and perturbed GHG concentrations and the control (fixed GHG concentrations) experiment. Thus, the estimates include (minimal) effects on the ERF from changes in land surface temperature because, unlike sea surface temperature, the land surface temperature is not prescribed (see Thornhill et al., (2021)). Likewise, the ERF of $N_2O$ may not

be accurate in the model. This problem may not be specific to IPSL-CM6A-LR: other climate models (e.g., CNRM-CM6) share the same radiative transfer code and most radiative transfer models used in some climate models have some degree of inaccuracy because they are designed to be computationally efficient (Collins et al., 2006; Fyfe et al., 2021; Pincus et al., 2016). Thus, there is a need to represent the ERF of $CH_4$ and $N_2O$ more accurately in order to better understand the effects of non-$CO_2$ GHGs mitigation on the Earth system. As developing better parameterizations of the gaseous optical properties is

beyond the scope of this study, we have developed an approach that adjusts $CH_4$ and $N_2O$ concentrations to 'effective' concentrations that generate $CH_4$ and $N_2O$ ERFs consistent with the reference estimates of IPCC AR6 (see Appendix A). The effective concentrations of $CH_4$ and $N_2O$ are used as input to the radiative transfer scheme of the climate model throughout the rest of this study but are referred to in the text and figures as the actual (equivalent) concentrations.

## 2.2 Experiment design

We perform and analyse a series of idealised global mean $CO_2$ and non-$CO_2$ concentration-driven ensemble experiments as summarised in Table 1 and Fig. 1. Including 50-year ramp-up, ramp-down, and stabilisation periods allows for the exploration of responses to increasing or decreasing $CO_2$ and non-$CO_2$ ($CH_4$ and $N_2O$) concentrations, as well as the long-term consequences and reversibility of their impacts on both the climate and carbon cycle. The inclusion of $CO_2$ and non-$CO_2$ concentration-driven experiments (with comparable ERF levels), a combined $CO_2$ and non-$CO_2$ concentration-driven

experiment, a biogeochemically coupled (BGC) experiment that includes only $CO_2$ physiological forcing (allowing $CO_2$ change to affect the carbon cycle of land and ocean), and a radiatively coupled (RAD) experiment that includes only $CO_2$ radiative forcing (not allowing $CO_2$ change to affect the carbon cycle), enables exploring the impacts of different forcing components on the climate and carbon cycle as well as potential feedback nonlinearities.



**Table 1. Description of experiments. Note that all experiments are analysed relative to their [piControl] counterparts.**

| Experiment name | Description | Maximum ERF* | Included carbon cycle feedback | Included carbon cycle terms from Eq.2 |
|---|---|---|---|---|
| [CO$_2$] | 0.7% CO$_2$ concentration increase per year from piControl for 50 years followed by 0.7% CO$_2$ decrease for 50 years. After CO$_2$ level returns to piControl, 50 years of stable piControl CO$_2$ concentrations. | 1.88 W m$^{-2}$ | Combined CO$_2$ β and CO$_2$ γ | $\Delta U_\beta$, $\Delta U_{\gamma,CO2}$, $\Delta U_{\beta\gamma,CO2}$ |
| [nonCO$_2$] | 2% CH$_4$ and 2% N$_2$O concentration increase per year from piControl for 50 years followed by 2% CH$_4$ and 2% N$_2$O decrease for 50 years. After CH$_4$ and N$_2$O levels return to piControl, 50 years of stable piControl CH$_4$ and N$_2$O concentrations. | 1.83 W m$^{-2}$ | Non-CO$_2$ γ | $\Delta U_{\gamma,nonCO2}$ |
| [CO$_2$ + nonCO$_2$] | Combined [CO$_2$] and [nonCO$_2$]. | 3.69 W m$^{-2}$ | Combined CO$_2$ β, CO$_2$ γ and non-CO$_2$ γ | $\Delta U_\beta$, $\Delta U_{\gamma,CO2+nonCO2}$, $\Delta U_{\beta\gamma,CO2+nonCO2}$ |
| [CO$_2$bgc] | Biogeochemically coupled [CO$_2$]. | | CO$_2$ β | $\Delta U_\beta$** |
| [CO$_2$rad] | Radiatively coupled [CO$_2$]. | | CO$_2$ γ | $\Delta U_{\gamma,CO2}$ |
| [CO$_2$bgc + nonCO$_2$] | [CO$_2$bgc] and [nonCO$_2$] combined. | | Combined CO$_2$ β and non-CO$_2$ γ | $\Delta U_\beta$, $\Delta U_{\gamma,nonCO2}$, $\Delta U_{\beta\gamma,nonCO2}$ |
| [CO$_2$] – [CO$_2$bgc] | Combination for comparison with [CO$_2$rad] | | CO$_2$ γ | $\Delta U_{\beta\gamma,CO2}$ |
| [CO$_2$bgc + nonCO$_2$] – [CO$_2$bgc] | Combination for comparison with [nonCO$_2$] | | Non-CO$_2$ γ | $\Delta U_{\beta\gamma,nonCO2}$ |
| [CO$_2$] + [nonCO$_2$] | Combination for comparison with [CO$_2$ + nonCO$_2$] | | Non-CO$_2$ γ and combined CO$_2$ β and CO$_2$ γ | $\Delta U_\beta$, $\Delta U_{\gamma,CO2}$, $\Delta U_{\beta\gamma,CO2}$, $\Delta U_{\gamma,nonCO2}$ |

*according to equations by Etminan et al. (2016)

**assuming negligible $\Delta U_{\gamma,CO2physiological}$ and $\Delta U_{\beta\gamma,CO2physiological}$

The experiment design uses a fixed land cover and constant (other than CO$_2$, CH$_4$ and N$_2$O) GHG and aerosol forcings that

might otherwise interfere with the interpretation of the results. The maximum ERF in our experiments is 3.69 W m$^{-2}$, which

corresponds to (actual) CO$_2$ concentration of 403 ppm, CH$_4$ concentration of 2175 ppb and N$_2$O concentration of 735 ppb, as

estimated from the equations by Etminan et al. (2016) (see also Appendix A). This ERF level (very much in line with the

current CO$_2$ concentration level of ca. 420 ppm) makes our experiments and results relevant to mitigation efforts in the near

future. The small differences in ERF between the [CO$_2$] and [nonCO$_2$] experiments are not significant when considering ramp-

up, ramp-down and full periods at $p < 0.05$.



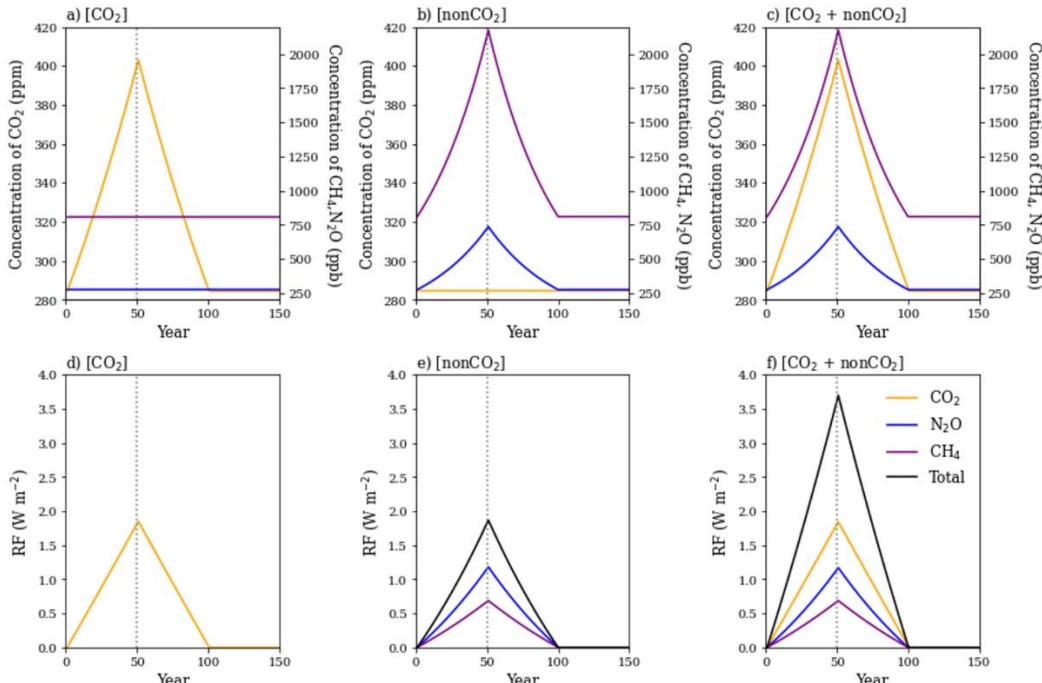

**Figure 1: Time series of input (a-c) CO₂, CH₄ and N₂O concentrations and (d-f) their respective radiative ERFs according to Etminan et al. (2016) equations for (a, d) [CO₂], (b, e) [nonCO₂] and (c, f) [CO₂ + nonCO₂] experiments. Note the different scales on the *y*-axes in a-c panels for CO₂ concentrations in ppm (left) and other GHG concentrations in ppb (right).**

We investigate $CO_2$ and non-$CO_2$ impacts on the climate by looking at the difference between [CO₂] and [nonCO₂] experiments and [CO₂rad] and [nonCO₂] experiments, hereafter referred to as [CO₂] – [nonCO₂] and [CO₂rad] – [nonCO₂], respectively. The experiments simultaneously manipulate $CH_4$ and $N_2O$ concentrations because our primary focus is to compare the effects of $CO_2$ with those of non-$CO_2$ gases (i.e., $CH_4$ and $N_2O$ collectively) in this study.

Three ensemble members are branched from years 1870, 2020, and 2170 of the CMIP6 piControl experiment, hereafter

[piControl]. We estimate the changes relative to the corresponding [piControl] periods in order to avoid the effects of low-frequency internal climate variability from the piControl (Fig. S1), as discussed in Bonnet et al. (2021). When reporting carbon sink/source in the following sections, we refer to the fluxes relative to the [piControl]. For diagnosing Atlantic Meridional Overturning Circulation (AMOC), we utilised the ocean overturning mass streamfunction in depth space (*msftyz* variable in CMIP6). Specifically, we calculated the maximum annual mean value of the streamfunction in the Atlantic basin north of

20°N through all model's depth layers (up to ca. 5800 m).

### 2.3 Carbon cycle feedback attribution

We examine the effects of $CO_2$ and non-$CO_2$ GHG on changes in carbon uptake (ΔU) using the well-established carbon cycle feedback framework, which involves estimating the carbon-concentration (β) and carbon-climate (γ) feedback parameters (Gregory et al., 2009; Williams et al., 2019), using Eq. (1):




$$\Delta U = \beta \times \Delta C_{CO2} + \gamma \times \Delta T + \varepsilon \,. \tag{1}$$

The $\beta$ parameter (GtC ppm$^{-1}$) refers to the changes in the carbon pools of land and ocean in response to changes in the atmospheric $CO_2$ concentration ($\Delta C_{CO2}$) and the $\gamma$ parameter (GtC K$^{-1}$) refers to the changes in the carbon pools in response to

changes in surface temperature ($\Delta T$), and $\varepsilon$ is a nonlinearity residual term. It can be estimated by making linear combinations of the various experiments. The atmospheric $CO_2$-induced $\gamma$ feedback can be derived from the [CO$_2$rad] - [piControl], as well as from the difference between the [CO$_2$] and [CO$_2$bgc] experiments, hereafter referred to as [CO$_2$] – [CO$_2$bgc]. The non-CO$_2$-driven carbon-climate feedback can be derived from the [nonCO$_2$] – [piControl] but also from the difference between the [CO$_2$ + nonCO$_2$] and [CO$_2$] experiments, hereafter referred to as [CO$_2$ + nonCO$_2$] – [CO$_2$], as well as from the difference between

the [CO$_2$bgc + nonCO$_2$] and [CO$_2$bgc] experiments, hereafter referred to as [CO$_2$bgc + nonCO$_2$] – [CO$_2$bgc] (Table 1). Since these experiments are intentionally designed to minimise differences in respective ERF levels, they mainly differ in terms of their impacts on the carbon cycle, particularly in relation to the inclusion or absence of the $\beta$ feedback.

In order to investigate the associated carbon-climate feedback nonlinearities, we propose the following theoretical framework. The changes in carbon uptake $\Delta U(\Delta C_{co2}, \ \Delta T)$ (GtC year$^{-1}$) in the fully coupled simulation can be defined as a function of changes

in $CO_2$ concentration ($\Delta C_{CO2}$) and temperature ($\Delta T$) that drives the temperature change. The formulation can be expanded to a Taylor series up to the second-order terms:

$$\Delta U = \Delta U(\Delta C_{co2}, \ \Delta T) = \frac{\partial U}{\partial C_{CO2}}\Delta C_{co2} + \frac{\partial U}{\partial T}\Delta T + \frac{1}{2}\frac{\partial^2 U}{\partial C_{CO2}\,\partial T}\Delta T\Delta C_{co2} + \frac{1}{2}\frac{\partial^2 U}{\partial C_{CO2}{}^2}(\Delta C_{co2})^2 + \frac{1}{2}\frac{\partial^2 U}{\partial T^2}(\Delta T)^2 + Res. \tag{2}$$

where $\Delta C_{CO2}$ and $\Delta T_{forc.}$ are the respective increments of $CO_2$ concentration and temperature relative to [piControl]. The higher-order terms are defined as a residual ($Res.$). We found them to be negligible in our case. We can disentangle the first- and second-order terms of the right-hand side of the Eq. (2) into terms that are purely dependent on $CO_2$ ($\Delta U_\beta$ ), on temperature ($\Delta U_\gamma$) and the cross term ($\Delta U_{\beta\gamma}$), as follows.

$$\Delta U_\beta = \frac{\partial U}{\partial C_{CO2}}\Delta C_{CO2} + \frac{1}{2}\frac{\partial^2 U}{\partial C_{CO2}{}^2}(\Delta C_{co2})^2 + Res., \tag{3}$$

$$\Delta U_\gamma = \frac{\partial U}{\partial T}\Delta T + \frac{1}{2}\frac{\partial^2 U}{\partial T^2}\Delta T^2 + Res., \tag{4}$$

$$\Delta U_{\beta\gamma} = \frac{1}{2}\frac{\partial^2 U}{\partial C_{CO2}\,\partial T}\Delta T\Delta C_{CO2} + Res., \tag{5}$$

The carbon-concentration $\beta$ feedback term $\Delta U_\beta$ may be estimated from [CO$_2$bgc] - [piControl], under the assumption that the

physiological $CO_2$ warming and its impacts on the carbon cycle are negligible. The carbon-climate $\gamma$ feedback terms for $CO_2$





($\Delta U_{\gamma,CO2}$) and non-CO$_2$ ($\Delta U_{\gamma,nonCO2}$) gases are estimated from [CO$_2$rad] - [piControl], and [nonCO$_2$] - [piControl], respectively. The difference between these terms ($\Delta U_{\gamma,CO2} - \Delta U_{\gamma,nonCO2}$) yields the difference between impacts of CO$_2$ and non-CO$_2$ forcing on the carbon-climate feedback. Finally, the cross term $\Delta U_{\beta\gamma}$, previously referred to as nonlinearity terms (Arora et al., 2020; Gregory et al., 2009; Schwinger and Tjiputra, 2018; Williams et al., 2019), may be estimated by utilising

a combination of experiments. The combination [CO$_2$] – [CO$_2$bgc] - [CO$_2$rad] gives $\Delta U_{\beta\gamma,CO2}$, and the combination [CO$_2$bgc + nonCO$_2$] – [CO$_2$bgc] - [nonCO$_2$] gives $\Delta U_{\beta\gamma,nonCO2}$). Analogously, the difference between these cross terms ($\Delta U_{\beta\gamma,CO2} - \Delta U_{\beta\gamma,nonCO2}$) yields the difference between the CO$_2$ and non-CO$_2$ forcings on the cross term of the carbon cycle feedback.

## 3 Results and discussion

### 3.1 Climate impacts

The analysis of global climate variables as a function of CO$_2$ concentration and GSAT shown in Fig. 2 follows a previous study by Boucher et al. (2012). Consistent with their findings, our results show that GSAT change lags behind GHG forcing by up to a decade. The lag increases with the increase in the forcing magnitude, so that the largest lag is in the [CO$_2$ + nonCO$_2$] experiment. Even after a ramp-down period and 50 years of constant GHG forcing at [piControl] levels, GSAT does not return to preindustrial values in all experiments (Fig. 2a). This can be explained by the inertia of the climate system, apparent in the

changes in the ocean heat uptake (OHU). The OHU increases during the ramp-up and decreases during the ramp-down period, being positive, i.e., taking up energy away from the atmosphere, during the ramp-up and first half of the ramp-down period. OHU turns negative by the end of the ramp-down and stays negative during 50 years of the stabilisation period, releasing back energy to the atmosphere (Fig. 2c–f). The hysteresis of the climate system is evident from the nearly linear relationship between maximum GSAT and mean GSAT during the stabilisation period (Fig. S2). The thermostatic (unrelated to ice sheet melting)

sea level increases in all experiments except for [CO$_2$bgc] and is closely related to the OHU. It does not recover (i.e. it is irreversible) under considered timescales (Fig. 2 k–n). The AMOC decreases with GSAT (with maximum reached for GSAT = 2 °C under [CO$_2$ + nonCO$_2$]), but fully recovers (Fig. S1b).





**Figure 2: Globally- and annually-averaged changes in model climate variables as a function of (a, c, g, k) time (year), (b, d, h, l) CO₂ concentration (ppm) / CH₄ concentration (ppb, only for [nonCO2]), (c, e, i, m) GSAT (ºC) and (f, j, n) cumulative GSAT (ºC*year) for (a, b) GSAT (ºC), (c–f) ocean heat uptake (W m⁻²), (g–j) cloud net radiative forcing (W m⁻²), and (k–n) thermosteric sea level change (m) under selected scenarios. The ramp-up, ramp-down and stabilisation periods are indicated by different line styles in all panels, except for those that have cumulative GSAT in horizontal axis. Thick lines indicate the ensemble means and thin lines correspond to three ensemble members.**

The CO₂ physiological warming that can be quantified by comparing [CO₂bgc] with [piControl] is not significant at a global

scale, when looking at 50-year ramp-up and ramp-down periods (green line in Fig. 2). The difference becomes significant





when considering 50-year stabilisation period at $p < 0.05$. Spatially, the differences are evident already during the ramp-up period, globally over land. Significant $CO_2$ physiological warming persists over Eurasia during the ramp-up period, and over the high latitudes of land and ocean during stabilisation period (Fig. S3a).

When comparing $CO_2$- and non-$CO_2$-induced radiative forcing ([$CO_2$rad] and [non$CO_2$] experiments) at a global scale, the non-$CO_2$ forcing leads to a lower GSAT peak and a slightly lower peak of thermostatic sea level rise compared to the $CO_2$ radiative forcing (brown and black lines of Fig. 2a, significant difference at $p < 0.05$). This cannot be explained just by a slightly higher ERF of the [$CO_2$rad] compared to [non$CO_2$] experiment (Table 1, Fig. 2a). Our results are consistent with Nordling et al. (2021) who show the higher sensitivity to non-$CO_2$ forcing compared to $CO_2$ forcing (Figure S4).

The combined effect of $CO_2$ physiological and radiative forcing leads to more warming in the coupled [$CO_2$] experiment compared to both the [$CO_2$rad] experiment. This is explained by the smaller negative cloud radiative feedback and the nonlinear impacts of $CO_2$ physiological warming, which amplifies the total $CO_2$ radiative warming (Fig. 2a and k). Spatially, by the end of the ramp-up period, the $CO_2$-induced total surface warming is larger than $CO_2$-induced radiative warming almost everywhere, except for the high northern latitudes over the land and ocean (Fig. S3). Our analysis indicates that this may be

due to a larger snow-cover driven albedo in the [$CO_2$] compared to the [$CO_2$rad], especially in the boreal spring (Fig. S5). These differences are also apparent during the ramp-down and stabilisation periods, revealing the scenario dependence of the surface temperature change (Tachiiri, 2020).

The lag in GSAT decrease is dominated by the high latitudes in all experiments. It can be partly explained by the polar amplification and albedo effects due to the "greening" of the northern high latitudes, i.e., increase in the interactive leaf area

index (LAI) (Figs. S5-6, see also Piao et al. (2020)). We found a difference in the land evapotranspiration between the $CO_2$ and non-$CO_2$ forcings, likely initiated by the physiological $CO_2$-driven decrease in land evapotranspiration (Fig. S7). The $CO_2$-driven decrease in the land evapotranspiration has been discussed elsewhere before (Zarakas et al., 2020). The divergence mentioned above can also be attributed to the $CO_2$-driven decrease in stomatal conductance over the $CO_2$-driven increase in LAI. Since this study includes only one ESM and three ensemble members, the robustness of these findings should be validated

by future studies.

### 3.2 Carbon cycle feedback

### 3.3 Carbon-concentration feedback

The presented (in Methods) theoretical framework enables attributing the γ carbon-climate feedback terms under the climate change driven by total $CO_2$ and non-$CO_2$ forcing, as well as in the presence and absence of the β carbon-concentration feedback

(Figs. 3, 4 and Table 2). The β feedback can be determined using the term $\Delta U_{\beta,CO2}$ (Eqs. 2, 3). It is accounted for in the $CO_2$ concentration-driven flux (i.e., β can be determined using the [$CO_2$bgc] experiment). Here we assume that $\Delta U_{\gamma,CO2physiological}$ and $\Delta U_{\beta\gamma,CO2physiological}$ are also negligible (see Section 3.1). An additional simulation using an offline land surface model would be required to quantify these terms more precisely.




**Table 2. Cumulative CO₂ and climate change-driven changes in the land and ocean carbon fluxes (GtC), shown as three- member**
**ensemble mean. The ± indicates one standard deviation among the three members. Note that all experiments are analysed relative**
**to their [piControl] counterparts.**

| Experiment | Terms | Years 1-50 (ramp-up) | | Years 51-100 (ramp-down) | | Years 101-150 (stabilisation) | | Total | |
|---|---|---|---|---|---|---|---|---|---|
| | | Land | Ocean | Land | Ocean | Land | Ocean | Land | Ocean |
| [CO₂bgc] | $\Delta U_{\beta,CO2}$ | 179.3±2.2 | 103.8±0.7 | -16.3±6.0 | -19.7±1.0 | -106.3±0.4 | -32.1±0.8 | 59.4±1.8 | 53.4±0.9 |
| [CO₂rad] | $\Delta U_{\gamma,CO2}$ | -18.6±2.4 | -2.4±0.2 | 4.5±5.7 | 0.0±1.4 | 11.4±4.0 | 0.3±1.9 | -2.2±2.0 | -2.1±0.6 |
| [nonCO₂] | $\Delta U_{\gamma,nonCO2}$ | -14.8±3.6 | -2.3±1.0 | 1.3±6.0 | -0.5±1.3 | 10.2±2.4 | 1.3±0.5 | -2.5±2.9 | -1.6±0.2 |
| [CO₂] – [CO₂bgc] | $\Delta U_{\gamma,CO2} + \Delta U_{\beta\gamma,CO2}$ | -14.7±1.0 | -4.5±0.2 | 3.8±3.0 | -1.4±1.0 | 8.2±1.9 | 0.6±0.5 | -3.8±3.1 | -5.5±1.4 |
| | $\Delta U_{\beta\gamma,CO2}$ | 2.9±2.1 | -1.6±0.4 | -0.5±7.3 | -1.1±0.5 | -2.4±5.9 | 0.2±2.3 | -1.2±5.0 | -2.6±2.0 |
| | $\Delta U_{\beta\gamma,nonCO2}$ | 2.6±1.5 | -1.2±1.0 | 2.1±7.9 | 0.1±1.1 | -1.2±1.5 | -0.8±1.3 | 1.5±4.0 | -1.9±1.0 |
| | $\Delta U_{\gamma,CO2} - \Delta U_{\gamma,nonCO2}$ | -3.8±5.0 | -0.1±0.9 | 3.2±3.6 | 0.5±0.9 | 1.2±2.6 | -1.0±2.2 | 0.3±2.7 | -0.5±0.7 |
| | $\Delta U_{\beta\gamma,CO2} - \Delta U_{\beta\gamma,nonCO2}$ | 0.4±3.5 | -0.6±0.7 | -3.5±3.4 | -1.6±1.0 | -1.6±7.4 | 1.3±3.3 | -3.6±5.5 | -1.0±2.8 |

The term $\Delta U_{\beta,CO2}$ dominates the changes over the land and ocean through all three considered periods (Fig. 3, Table 2). Since the maximum surface warming levels of all considered experiments are below 2 °C, the land and ocean carbon fluxes are primarily controlled by the β feedback, which induces carbon sink over land and ocean during the ramp-up period (Fig. 3 a-d). Yet, the contribution of β is expected to decrease with higher warming (Arora et al., 2020; Melnikova et al., 2023). Nearly two-thirds of land and half of ocean carbon accumulated during the ramp-up period due to the atmospheric CO₂ increase is being released during the latter periods, which is consistent with previous studies (Asaadi et al., 2024; Chimuka et al., 2023). Ocean behaves as an enhanced atmospheric CO₂ concentration-driven carbon sink in all regions during the ramp-up period (Fig. 4). However, during the ramp-down, β induces a carbon source over all ocean regions, except for the Atlantic and Southern Oceans. It drives a carbon source over all regions during the stabilisation period.

The behaviour of the β feedback over land can be better explained by analysing the gross primary production (GPP) and the autotrophic and heterotrophic respiration (Ra and Rh) fluxes. Land GPP, representing photosynthetic uptake, increases during the ramp-up period under elevated CO₂ concentration and decreases almost linearly with decreasing CO₂, showing only a small hysteresis (Fig. S9). In contrast, both autotrophic and heterotrophic respiration (Ra and Rh) exhibit a larger hysteresis, which leads to a prolonged period of the carbon release to the atmosphere. This finding differs from the results of a recent study based on intermediate-complexity ESM, where authors did not detect any hysteresis in Rh (see Fig. S1 of Chimuka et al. (2023)). Our results suggest that while there may be initial carbon sequestration benefits gained during elevated CO₂ periods, these benefits are susceptible to being lost as CO₂ concentrations decline due to decreased photosynthesis and increased respiration, albeit at a reduced rate.





**Figure 3. Global cumulative carbon fluxes (GtC) over (a, b, e, f, i) land and (c, d, g, h, j) ocean as a function of (a, c, e, g, i, j) time (year), (b, d) CO₂ concentration (ppm) and (f, h) GSAT changes (ºC) under selected scenarios. The ramp-up, ramp-down and stabilisation periods are indicated by different line styles. Thick lines indicate the ensemble means and thin lines correspond to other ensemble members.**

The spatial variation of cumulative net carbon uptake provides further details on the feedback behaviour (Figs. 4, S10-S12). During the CO₂ ramp-up phase, β feedback triggers a land carbon sink in all regions. However, during the ramp-down phase, it induces a net carbon source over subtropical regions while still driving a land carbon sink in northern high-latitudes, so that global land becomes a carbon source in the middle of the ramp-down phase. Finally, during the stabilisation period, β feedback causes all land regions to transition into net carbon source.

Subtropical and southern land regions exhibit a shorter hysteresis in response to decreasing CO₂ concentrations. This disparity arises from the larger proportion of carbon accumulated in aboveground vegetation biomass in southern regions, contrasting with the greater fraction stored in soils within northern latitudes (Figs. S11-S13). The prolonged duration of β benefits in northern mid- to high latitudes is attributed to the longer carbon turnover time, particularly in soils, compared to tropical regions (Fig. S10).



### 3.4 Carbon-climate feedback

The carbon-climate γ feedback involves two components, namely $\Delta U_\gamma$ and the cross term $\Delta U_{\beta\gamma}$ in Eq. 2. (see also Eqs. 4 and 5). Here we attribute the cross term to be part of γ feedback in order to keep the legacy of previous studies on the carbon cycle feedback framework (Arora et al., 2020; Schwinger and Tjiputra, 2018). The term $\Delta U_\gamma$ corresponds to the flux arising from

variations in radiative forcing (such as in [nonCO2] and [CO2rad] experiments). We find that $\Delta U_\gamma$ for $CO_2$ and non-$CO_2$ ($\Delta U_{\gamma,CO2}$ and $\Delta U_{\gamma,nonCO2}$, respectively) are equivalent under nearly equivalent levels of ERF (compare panels b and c of Fig. 4, see also $\Delta U_{\beta\gamma,CO2} - \Delta U_{\beta\gamma,nonCO2}$ in Table 2). The $\Delta U_{\beta\gamma}$ induces non-negligible differences between climate change-induced carbon flux when comparing experiments with presence or absence of atmospheric $CO_2$ concentration change, reaching ca. 15-18% of the land $\Delta U_\gamma$ and exceeding the ocean $\Delta U_\gamma$, cumulative over the ramp-up period (Table 2). During the ramp-up

phase, the cross term $\Delta U_{\beta\gamma}$ induced a decrease in γ-driven carbon source over land and increase in γ-driven carbon source over the ocean (Fig. 3). We find that the cross term $\Delta U_{\beta\gamma}$ is equivalent for $CO_2$ and non-$CO_2$ under similar levels of ERF for the land but not for the ocean (compare panels f and g of Fig. 4, see also $\Delta U_{\beta\gamma,CO2} - \Delta U_{\beta\gamma,nonCO2}$ in Table 2). Over the ocean, the cross-term differences for $CO_2$ and non-$CO_2$ forcing arise already during the ramp-up and propagate during the ramp-down and stabilisation phases, spatially concentrating in the deep mixing Southern Ocean area.

Over the land, the γ is positive in the mid- to high latitudes and negative in the tropical regions over the land (Figs. 4, S11-12, see also Melnikova et al. (2021)). During the ramp-up, γ drives carbon sink over the northern mid- to high- latitudes and carbon source over the subtropical regions and the Southern Hemisphere, with larger magnitude of changes in the experiments, in which $CO_2$ concentration change is present (Figs. 4, S11-13). Here the cross term $\Delta U_{\beta\gamma}$ makes more biomass over high latitudes available for a positive γ feedback (larger carbon sink). During the ramp-down, γ increases the carbon sink over high

latitudes and weakens the carbon source over the tropics. Here the term $\Delta U_{\beta\gamma}$ makes more biomass available globally for negative γ (larger carbon source), driven by the lagged (after GHG concentrations) temperature change. The differences (under the presence / absence of cross term) $\Delta U_{\beta\gamma}$ diminish for land but not for the ocean during the ramp-down and stabilisation periods.

The cross term $\Delta U_{\beta\gamma}$ results in a larger climate change-driven carbon source in the ocean (Fig. 3). Unlike over the land, the

cross term $\Delta U_{\beta\gamma}$ contribution increases during the ramp-down phases of considered GHG concentration scenarios. Previously, it was referred to as a nonlinearity of carbon cycle feedback, as discussed in detail by Schwinger and Tjiputra (2018). These authors warn that RAD experiments may underestimate the carbon-climate feedback (when compared to full-coupled - BGC experiments), because "the reduction of sequestration of preformed dissolved inorganic carbon under high atmospheric $CO_2$ is not taken into account." In this study, the interaction with β leads to 5% to 6% more climate change-driven carbon loss

(compared to when atmospheric $CO_2$ is constant) relative to the total net 150-year net carbon uptake under the [CO2] experiment. Spatially, although the Southern Ocean behaves as the largest ocean carbon sink in all considered experiments





involving atmospheric $CO_2$ changes, together with the Atlantic Ocean it experiences the largest regional ocean carbon loss driven by climate change (Fig. 4).



**Figure 4. Spatial variation of land and ocean carbon fluxes (GtC, negative to the atmosphere) cumulative over 50 years of (first column) ramp-up, (second column) ramp-down, (third column) stabilisation phases and (last column) full 150-year period. The data for three-member-ensemble mean are used.**




Our findings provide evidence on the effectiveness of non-CO$_2$ GHG mitigation. While it can effectively reduce GSAT peak, non-CO$_2$ GHG mitigation may also lead to smaller climate change-driven losses in the ocean carbon sink. In the real world,

the presence / absence of $\Delta U_{\beta\gamma}$ suggests disparities between CO$_2$ mitigation efforts (which involve carbon-concentration feedback alterations) and non-CO$_2$ mitigation efforts. The CO$_2$- and non-CO$_2$ concentration-driven climate change leads to unequal decrease in carbon uptake, especially apparent for the ocean on a global scale (Fig. 3). Reducing CO$_2$ concentrations for climate mitigation implies alteration of all three terms of the proposed carbon cycle feedback attribution framework, namely $\Delta U_\beta$, $\Delta U_\gamma$ and $\Delta U_{\beta\gamma}$. Reducing non-CO$_2$ GHG concentrations, such as CH$_4$ and N$_2$O, implies alteration of $\Delta U_\gamma$ term alone.

Reducing both CO$_2$ and non-CO$_2$ concentrations implies alteration all $\Delta U_\beta$, $\Delta U_\gamma$ and $\Delta U_{\beta\gamma}$ terms but with larger change in $\Delta U_\gamma$ and $\Delta U_{\beta\gamma}$ terms. From this point, combining CO$_2$ and non-CO$_2$ reduction measures may be more effective for climate change mitigation, compared to the CO$_2$ reduction measures alone. This finding should be confirmed with emission-driven experiments that consider GHGs atmospheric lifetimes.

### 3.5 Radiative forcing and carbon cycle feedback additivity

In order to overcome the small signal-to-noise ratio of the considered experiments as well as the regional differences in the radiative forcing between [CO$_2$] and [nonCO$_2$] experiments, we compare the (1) [CO$_2$ + nonCO$_2$] experiment that include both CO$_2$ and non-CO$_2$ effects with (2) the sum of two [CO$_2$] and [nonCO$_2$], that include CO$_2$ and non-CO$_2$ effects, accordingly (Figs. 5 and S14-15). The climate effects, defined via temperature change, differ during the ramp-up and ramp-down periods (Fig. S14). These differences imply non-additivity of radiative forcing and can also be attributed to biophysical feedback.

As for the carbon cycle feedback, the [CO$_2$ + nonCO$_2$] experiment that has all feedback is different from the sum of two experiments [CO$_2$] + [nonCO$_2$] both over the land and ocean (Fig. 5). The differences are larger and stay longer over the ocean. This implies non-additivity of carbon cycle feedback. From the proposed carbon cycle feedback attribution framework, the non-additivity arises from nonequality of ($\Delta U_{\gamma,CO2+nonCO2} + \Delta U_{\beta\gamma,CO2+nonCO2}$) and ($\Delta U_{\gamma,CO2} + \Delta U_{\beta\gamma,CO2} + \Delta U_{\gamma,nonCO2} + \Delta U_{\beta\gamma,nonCO2}$). The significant ($p < 0.1$) difference over the land is in the high-latitude region, where "all-effects together" [CO$_2$

+ nonCO$_2$] experiment yields a larger carbon sink during the ramp-up phase. The largest difference over the ocean is in the Southern Ocean, followed by the North Atlantic Ocean. The [CO$_2$ + nonCO$_2$] experiment has large carbon sink is in the Southern Ocean compared to the sum of two experiments. This might be related to the saturation of the decrease in the mixed layer depth with more warming, but a more thorough study is needed to confirm such phenomena.



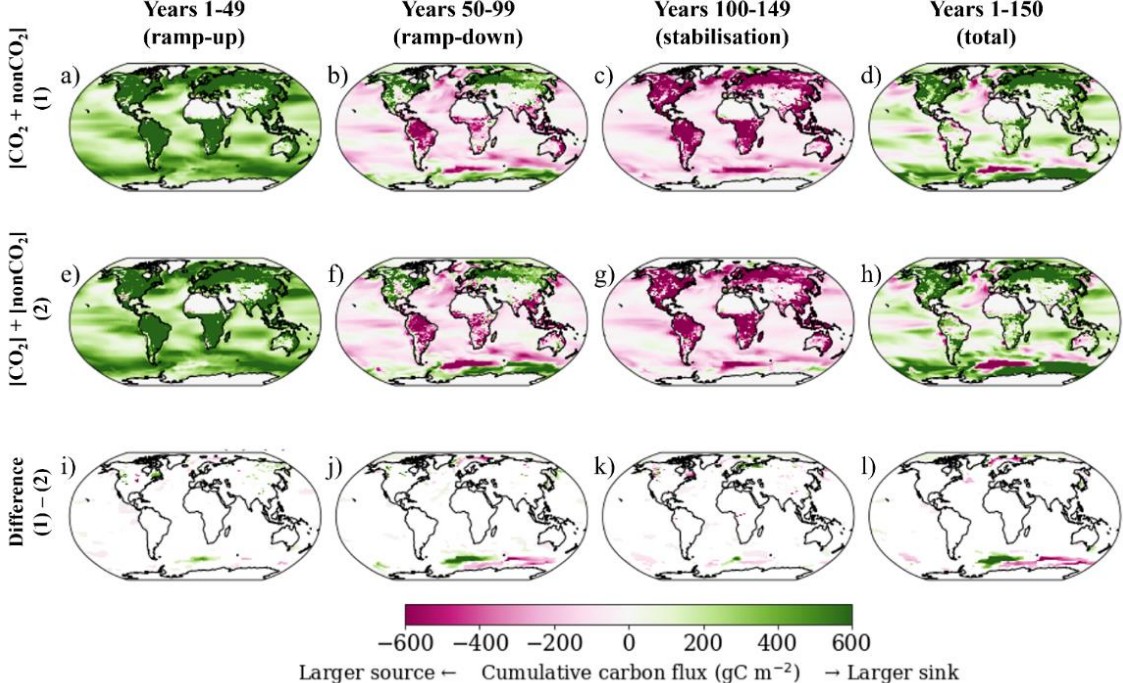

**Figure 5. Spatial variation of three-member-ensemble mean land and ocean carbon fluxes (GtC, negative to the atmosphere) cumulative over 50 years of (a, e, i) ramp-up, (b, f, j) ramp-down, (c, g, k) stabilisation phases and (d, h, l) full 150-year period. We draw only significantly different grids between (i-l) [CO₂ + nonCO₂] and [CO₂] + [nonCO₂] experiments using three ensemble members ($p < 0.1$ based on $t$-test, $N$=60).**

## 4 Limitations and future research directions

To our knowledge, this is the first study of its kind comparing idealised $CO_2$ and non-$CO_2$ ramp-up and ramp-down scenarios for their effects on global temperature change and carbon climate, carbon concentration feedback. Below we draw attention to the caveats and limitations that should be addressed in future studies.

First, since IPSL-CM6A-LR does not have interactive modules of the $CH_4$ and $N_2O$ cycles, the changes in stratospheric water vapour, aerosols, and tropospheric ozone due to atmospheric $CH_4$ changes, as well as the effects of nitrogen deposition on the

carbon cycle are not considered in this study. Future studies could consider simulations separately for $CH_4$ and $N_2O$.

Second, when interpreting the results, it should be kept in mind that some carbon cycle processes in IPSL-CM6A-LR such as permafrost and fire are not considered. Yet, these should have only a limited impact on our results, given the (relatively) small warming levels in the considered experiments. Previous studies have shown that IPSL-CM6A-LR estimates one of the smallest soil carbon pools among CMIP6 models, which may lead to an underestimation of the carbon-climate feedback (Arora et al.,

2020; Melnikova et al., 2021).





Third, the results of the present study are limited by the use of a single ESM and a small number of ensemble members. Conducting similar experiments with other ESMs and using larger ensemble runs, which are particularly valuable in the low warming scenarios, as well as complementing the findings of our study with emission-driven experiments, could contribute to validating and extending our findings.

**5 Conclusions**

This study first presents a novel approach to recalibrate the ERF of $CH_4$ and $N_2O$ in ESMs without changing the radiative scheme of the model. We then discuss the effects of increases and decreases in the concentrations of the $CO_2$ and non-$CO_2$ GHGs on the surface climate and carbon cycle. We find only small differences between $CO_2$ and non-$CO_2$ ramp-up and ramp-down forcing on global and regional climate. We show that under the $CO_2$ concentration change, its physiological warming

adds to the $CO_2$ radiative warming, leading to a higher peak in GSAT.

The differences in climate responses can be linked to differences in the carbon cycle feedback. We show that $CO_2$- and non-$CO_2$-driven carbon-climate feedback are nearly equivalent at a global scale. However, the presence of carbon-concentration feedback amplifies the climate change-driven carbon loss, especially over the ocean. We propose a novel framework to disentangle the carbon-climate feedback into a component that is purely driven by climate change i.e., expressed as a

temperature term, and a component driven by climate change under the presence of the carbon-concentration feedback, i.e., a cross term. Since the cross term can be quantified from the difference between fully coupled, BGC and RAD simulations, we advocate for continuing to carry out all three types of experiments in the future phases of CMIP. We further warn that the cross term and non-additivity of feedback should be considered in the simple climate models (emulators).

Finally, this study showcases the additional benefits of non-$CO_2$ GHG mitigation on a smaller reduction of the ocean carbon

sinks under overshoot scenarios. We stress that our findings do not imply that non-$CO_2$ GHG mitigation should be given the priority to other means to mitigate climate change but provide an insight on the intricate interplay between the carbon-concentration and $CO_2$- and non-$CO_2$-driven carbon-climate feedback to inform comprehensive mitigation strategies.

**Appendix A**

A set of 40-year idealised IPSL-CM6A-LR simulations (840 years in total) has been carried out. In these experiments, the sea surface temperature (SST) and sea ice fractions were fixed to their preindustrial levels. $CH_4$ concentration levels were kept at $2{\times}CH_4$, $3{\times}CH_4$, $4{\times}CH_4$, $5{\times}CH_4$, $8{\times}CH_4$, and $12{\times}CH_4$, and $N_2O$ concentration levels were kept at $1{\times}N_2O$, $1.25{\times}N_2O$, $1.5{\times}N_2O$, and $1.75{\times}N_2O$ (Table A1).

**Table A1. Description of idealised recalibration experiments using IPSL-CM6A-LR**



| № | Name | Concentration | | |
|---|---|---|---|---|
| | | CH₄ (ppb) | N₂O (ppb) | CO₂ (ppm) |
| 0 | *piClim* | 808.25 | 273.02 | 284.32 |
| 1 | 2×CH₄ | 1616.50 | 273.02 | 284.32 |
| 2 | 3×CH₄ | 2424.75 | 273.02 | 284.32 |
| 3 | 4×CH₄ | 3233.00 | 273.02 | 284.32 |
| 4 | 5×CH₄ | 4041.25 | 273.02 | 284.32 |
| 5 | 8×CH₄ | 6466.00 | 273.02 | 284.32 |
| 6 | 12×CH₄ | 9699.00 | 273.02 | 284.32 |
| 7 | 1×CH₄-1d25N₂O | 808.25 | 341.28 | 284.32 |
| 8 | 2×CH₄-1d25N₂O | 1616.50 | 341.28 | 284.32 |
| 9 | 3×CH₄-1d25N₂O | 2424.75 | 341.28 | 284.32 |
| 10 | 4×CH₄-1d25N₂O | 3233.00 | 341.28 | 284.32 |
| 11 | 5×CH₄-1d25N₂O | 4041.25 | 341.28 | 284.32 |
| 12 | 8×CH₄-1d25N₂O | 6466.00 | 341.28 | 284.32 |
| 13 | 12×CH₄-1d25N₂O | 9699.00 | 341.28 | 284.32 |
| 14 | 1×CH₄-1d5N₂O | 808.25 | 409.53 | 284.32 |
| 15 | 2×CH₄-1d5N₂O | 1616.50 | 409.53 | 284.32 |
| 16 | 3×CH₄-1d5N₂O | 2424.75 | 409.53 | 284.32 |
| 17 | 4×CH₄-1d5N₂O | 3233.00 | 409.53 | 284.32 |
| 18 | 5×CH₄-1d5N₂O | 4041.25 | 409.53 | 284.32 |
| 19 | 8×CH₄-1d5N₂O | 6366.00 | 409.53 | 284.32 |
| 20 | 12×CH₄-1d5N₂O | 9699.00 | 409.53 | 284.32 |
| 21 | 1×CH₄-1d75N₂O | 808.25 | 477.79 | 284.32 |

Fig. A1 shows the set of 40-year time series of global mean radiative forcing based on 21 idealised experiments. The *piClim* experiment holds pre-industrial levels of $CO_2$, $CH_4$ and $N_2O$ concentrations. The mean interannual variation of the radiative forcings (one standard deviation) is 0.15 Wm$^{-2}$. The first 10 years of the experiments were dropped to allow the climate to adjust to the new radiative equilibrium after an abrupt change from the pre-industrial levels. The last 30 years were used to obtain eighteen data-points of mean global ERF by IPSL-CM6A-LR relative to the levels obtained from the *piClim* experiment.

The ERF is estimated as a TOA imbalance difference between each experiment and the *piClim*.



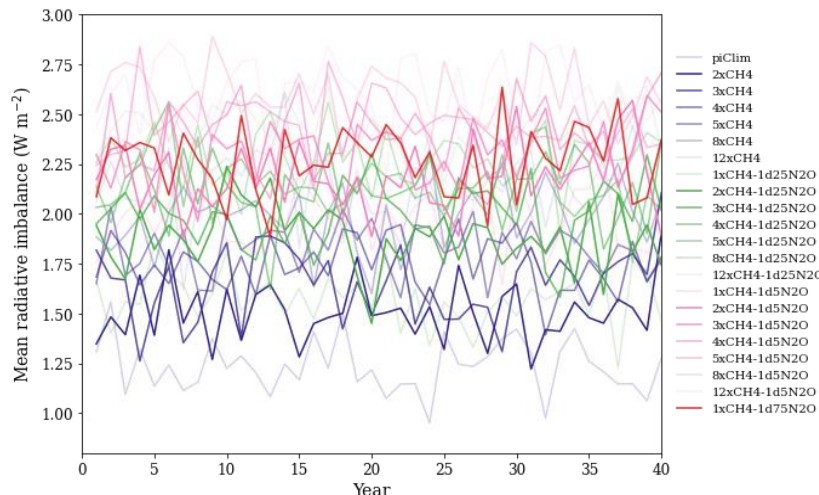

**Figure A1. Time series of global mean radiative imbalance of IPSL-CM6A-LR idealised experiments**

There are two frequently used sets of equations to derive radiative forcing of well mixed greenhouse gases, e.g., $CO_2$, $CH_4$ and $N_2O$, etc., based on their concentrations. The first set from Myhre et al. (Myhre et al., 1998), thereafter, M98, was used in IPCC AR3:

$$RF(CO2) = 5.35 \times ln\,(CO2/CO2_{t=0})$$

$$RF(CH4) = 0.036 \times (\sqrt{CH4} - \sqrt{CH4_{t=0}}) - (f(CH4, N2O_{t=0}) - f(CH4_{t=0}, N2O_{t=0}))$$

$$RF(N2O) = 0.12 \times \left(\sqrt{N2O} - \sqrt{N2O_{t=0}}\right) - (f(CH4_{t=0}, N2O) - f(CH4_{t=0}, N2O_{t=0}))$$

(A1)

$$f(CH4, N2O) = 0.47 \times ln[1 + 2.01 \times 10^{-5} \times (CH4 \times N2O)^{0.75} + 5.31 \times 10^{-15} \times CH4 \times (CH4 \times N2O)^{1.52}]$$

Here the $CO_2$, $CH_4$ and $N_2O$ indicate their concentrations, where the units are ppm, ppb, and ppb.

Etminan et al. (2016), thereafter, E16, improved the IPCC AR3 equations by inclusion of the shortwave (near-infrared) bands of $CH_4$. Their set of equations was used in IPCC AR6. The radiative forcing of $N_2O$ in the M98 equation depends on the $CH_4$ and $N_2O$ concentrations, while the radiative forcing of $N_2O$ in the equation by E16 depends on the $CO_2$, $CH_4$ and $N_2O$ concentrations:

$$RF(CO2) = [-2.4 \times 10^{-7} \times (CO2 - CO2_{t=0})^2 + 7.2 \times 10^{-4} \times |CO2 - CO2_{t=0}|$$
$$- 2.1 \times 10^{-4} \times \frac{1}{2}(N2O + N2O_{t=0}) + 5.36]\,ln\,(\frac{CO2}{CO2_{t=0}})$$

$$RF(CH_4) = \left[-1.3 \times 10^{-6} \times \frac{1}{2}(CH4 + CH4_{t=0}) - 8.2 \times 10^{-6} \times \frac{1}{2}(N2O + N2O_{t=0}) + 0.043\right] \times (\sqrt{CH4}$$
$$- \sqrt{CH4_{t=0}})$$

(A2)





$$RF(N_2O) = [-8.0 \times 10^{-6} \times \frac{1}{2}(CO2 + CO2_{t=0}) + (4.2 \times 10^{-6} \times \frac{1}{2}(N2O + N2O_{t=0})$$

$$- 4.9 \times 10^{-6} \times \frac{1}{2}(CH4 + CH4_{t=0}) + 0.117] \times (\sqrt{N2O} - \sqrt{N2O_{t=0}})$$

In the sets of Eqs. (A2), the radiative forcing due to $CH_4$ depends not only on the $CH_4$ concentration but also on that of $N_2O$ (and conversely) because $CH_4$ and $N_2O$ absorption bands overlap to some extent. These simplified equations are therefore additive: the radiative forcing due to a change in $CH_4$ and $N_2O$ concentrations ($CH_4$, $N_2O$ ) relative to reference (preindustrial) values ($CH4_{t=0}$, $N2O_{t=0}$) is equal to:

$RF(CH4, N2O) = RF_{CH4}(CH4, N2O) + RF_{N2O}(H4, N2O)$  (A3)

The effect of $CO_2$ on the radiative forcing of $N_2O$ is small (<5%), and thus, for simplicity, is neglected in the rest of this study (Fig. A2).

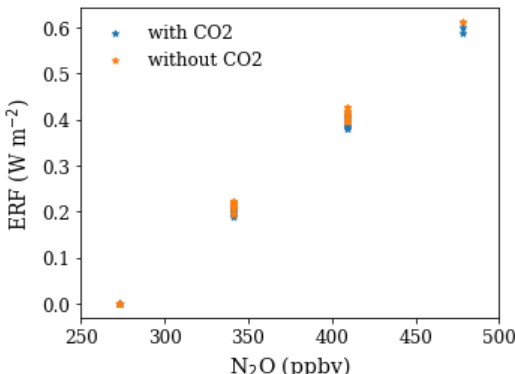

**Figure A2.The ERF estimated from E16 with and without accounting for CO₂ impact on the N₂O forcing (including/excluding the**
**term $-8.0 \times 10^{-6} \times 0.5 \times (CO_2^{real\, con-n} + CO_{2,t=0})$ in the equation) using the concentration values of the idealised experiments**
**described in the Table 1 with preindustrial CO₂ concentration (284.32 ppm) and the same set of the experiments with $3 \times CO_2$**
**concentration (852.96 ppm).**

Figure A3 shows the ERFs simulated by IPSL-CM6A-LR and those estimated from the two equations from the IPCC report and revised by E16. IPSL-CM6A-LR underestimates $CH_4$ ERF and overestimates $N_2O$ ERF, relative to both equations.

Considering the improvements introduced by E16 to include the shortwave bands of $CH_4$, we use their equations (and not of IPCC AR3) to recalibrate the IPSL-CM6A-LR model.



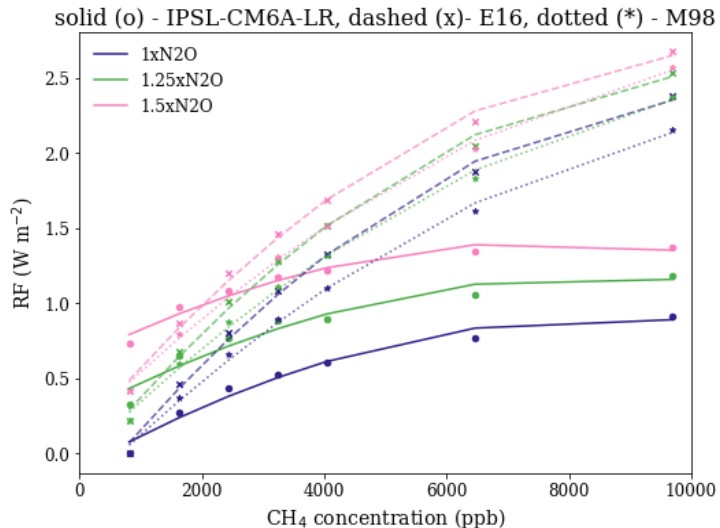

**Figure A3. The ERF of CH₄ and N₂O from IPSL-CM6A-LR idealised experiments (points and fitted solid lines) and estimated from M98 (\* and fitted dotted lines) and E16 (x and fitted dashed lines) equations, fitted to polynomial regressions for three levels of N₂O concentrations.**


A system of equations can convert the input CH₄ and N₂O concentrations to the effective concentrations so that, if used in the climate model, would yield actual forcing from the equations by E16. Among several linear and nonlinear functions to relate the actual concentrations of CH₄ and N₂O with the concentrations seen by IPSL-CM6A model (effective concentrations) that have been tested, the following set of equations yielded the best fit:

$$CH_4^{effective} = CH4_{t=0} + a + \times (CH4^{actual} - CH4_{t=0})^c \; N2O^{effective}$$

$$N2O^{effective} = N2O_{t=0} + b \times (N2O^{actual} - N2O_{t=0})^d$$

(A4)


The initial values of CH₄ and N₂O concentrations are fixed at the preindustrial levels ($t = 0$). Those of the effective CH₄ and N₂O concentrations are also assumed to take the same respective preindustrial levels. The Eq. (A1) is used to relate the effective concentrations of CH₄ and N₂O of IPSL-CM6A-LR to the ERF estimated by E16:

$$RF_{Etm} = RF_{CH4, Etm} + RF_{N2O, Etm}$$

$$= \left( -1.3 \times 10^{-6} \times \frac{1}{2}(CH4^{effective} + CH4_{t=0}) - 8.2 \times 10^{-6} \times \frac{1}{2}(N2O^{effective} + N2O_{t=0}) \right.$$

$$\left. + 0.043 \right) \times (\sqrt{CH4^{effective}} - \sqrt{CH4_{t=0}}) + (4.2 \times 10^{-6} \times \frac{1}{2}(N2O^{effective} + N2O_{t=0})$$

$$- 4.9 \times 10^{-6} \times \frac{1}{2}(CH4^{effective} + CH4_{t=0}) + 0.117) \times (\sqrt{N2O^{effective}} - \sqrt{N2O_{t=0}})$$

(A5)




We optimise the four parameters of Eq. (A4) by minimising the sum of squared residuals between ERF estimated by E16 equations and simulated by IPSL-CM6A-LR using a Python implementation of the Limited-memory Broyden–Fletcher–Goldfarb–Shanno bound-constrained algorithm (L-BFGS-B) that is an algorithm for solving large nonlinear optimization problems with simple bounds (Byrd et al., 1995). The cost function is defined as:

$$CF = \sum_{i}^{i=21}(RF_{IPSL} - RF_{Etm})^2 \tag{A6}$$

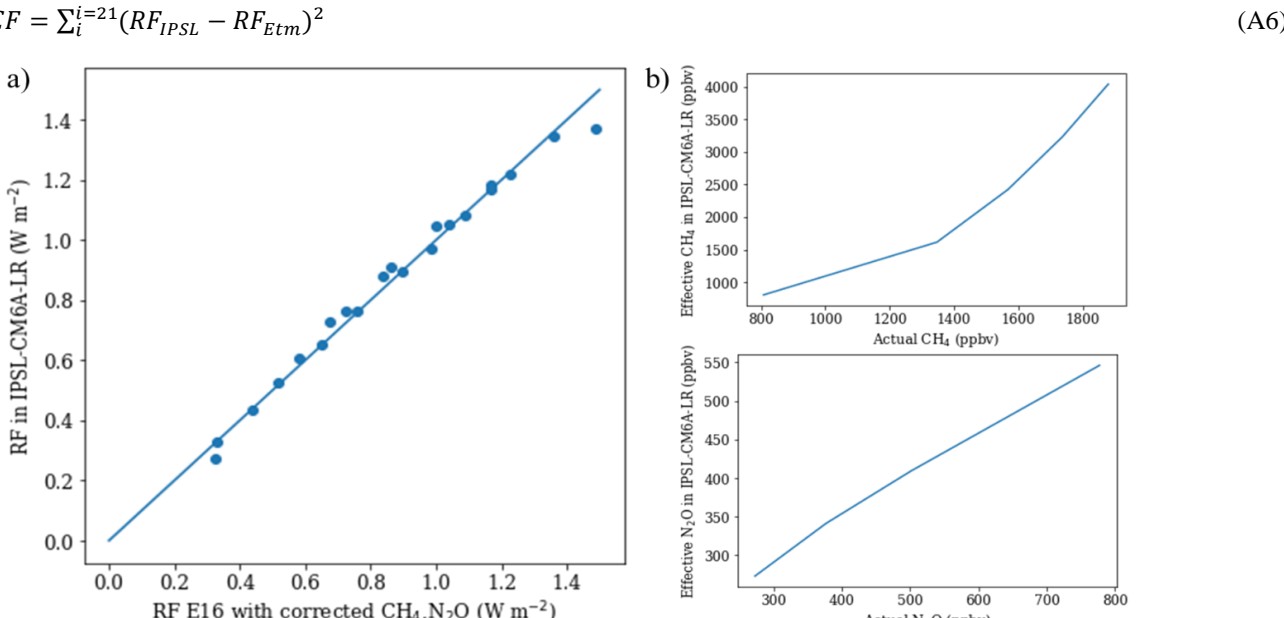

**Figure A4. (a) Scatterplot of ERF simulated by IPSL-CM6A and estimated by E16 equations using the corrected effective concentrations from Eq. (A3), and (b) The effective IPSL-CM6A-LR concentrations of $CH_4$ and $N_2O$ as a function of the actual concentrations derived via a system of nonlinear functions.**

The residual error after fit equals 0.03 $(Wm^{-2})^2$ (Fig. A4). The solution of the cost function provides the estimates of four uncertain parameters for Equations 3, indicated in Table A2. The effective IPSL-CM6A-LR concentrations of $CH_4$ and $N_2O$ are functions of the actual concentrations (Fig. A4b). Higher effective $CH_4$ and lower $N_2O$ effective concentrations are needed for IPSL-CM6A-LR to reproduce the ERF in agreement with IPCC estimates.

**Table A2. Estimated parameters in Eq. (A4)**

| Parameter | Estimate |
|---|---|
| a | 19.44701978 |
| b | 0.84856644 |
| c | 0.49593024 |
| d | 1.13865386 |

The estimated parameters were applied to derive the effective IPSL-CM6A-LR concentrations for a target scenario (Fig. A5). Higher $CH_4$ and lower $N_2O$ concentrations are required to reproduce the ERF correctly.





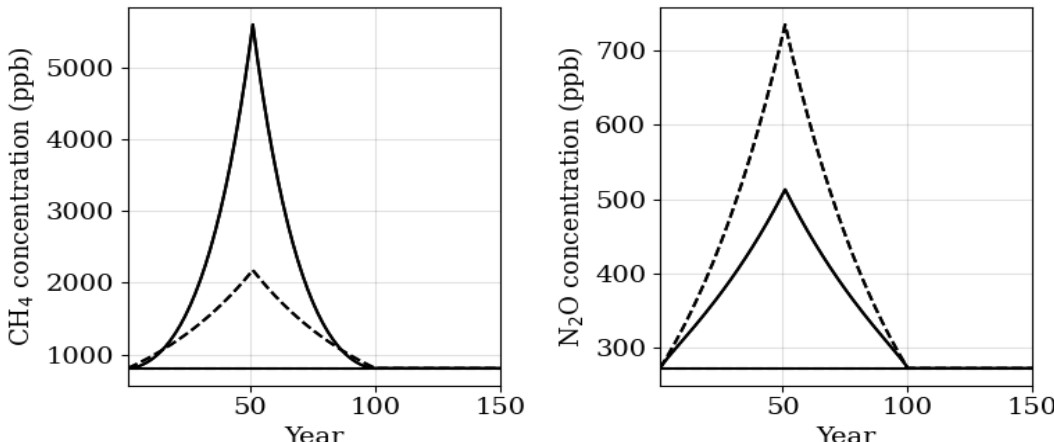

**Figure A5. The actual (dashed lines) and new recalibrated effective CH₄ and N₂O IPSL-CM6A-LR concentrations (solid lines) for the CH₄-N₂O experiment of this study.**

**Data and materials availability**

The piControl output data from the CMIP6 simulations are available from the CMIP6 archive: https://esgf-node.llnl.gov/search/cmip6 (WCRP, 2022)), additional IPSL-CM6A-LR outputs are available upon request. The Jupyter

notebook and data to reproduce the figures will be stored in the Zenodo archive upon publication.

**Acknowledgments**

The IPSL-CM6 simulations were performed using the HPC resources of TGCC under the allocation 2022-A0120107732 and 2023-A0140107732 (project gencmip6) provided by GENCI (Grand Equipement National de Calcul Intensif). This work benefited from the technical support and discussions with Adriana Sima and Nicolas Lebas at IPSL, Arnaud Caubel and

Sébastien Nguyen at LSCE (France) and Michio Kawamiya at JAMSTEC (Japan).

**Funding**

This work benefited from State assistance managed by the National Research Agency in France under the Programme d'Investissements d'Avenir under the reference ANR-19-MPGA-0008. We further acknowledge the Program for the Advanced Studies of Climate Change Projection (SENTAN, grant number JPMXD0722681344) from the Ministry of Education, Culture,

Sports, Science and Technology, Japan. We acknowledge the European Union's Horizon Europe research and innovation program under Grant Agreement N° 101056939 (RESCUE – Response of the Earth System to overshoot, Climate neUtrality and negative Emissions) and N° 101081193 (OptimESM – Optimal High Resolution Earth System Models for Exploring



Future Climate Changes). P.C. acknowledges support from the CALIPSO (CArbon Losses in Plants, Soils and Ocean) project funded through the generosity of Eric and Wendy Schmidt by recommendation of the Schmidt Futures program.

**Author contributions**

IM and OB initiated the study. OB, Ktan and IM developed methodology and study ideas. IM and OB performed IPSL-CM6A-LR simulations. IM performed data analysis and wrote the initial draft. All authors contributed to the discussions, writing, and revising the paper.

**Competing interests**

Authors declare that they have no competing interests.

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
