# Peer review of "Carbon cycle and climate feedbacks under CO2 and non-CO2 overshoot pathways"

_EGUsphere, 2024_

## Referee Comment (RC2)

**Review for Carbon Cycle and Climate Feedback under $CO_2$ and non-$CO_2$ Overshoot Pathways by Melnikova et al., submitted to Earth Systems Dynamics (EGUsphere)**

This study compares the climate and carbon cycle response to equivalent $CO_2$ and non-$CO_2$ forcings using a set of idealized concentration-driven simulations. The authors find that the climate-carbon feedback is dominant under non-$CO_2$ forcing whereas both the carbon-concentration and climate-carbon feedbacks are important under the $CO_2$ forcing. Under both $CO_2$ and non-$CO_2$ forcings, the land and ocean carbon uptake due to both feedbacks is quantified along with a cross term i.e., a term that quantifies the response to climate change in the presence of $CO_2$ concentration.

The manuscript reads well – the introduction and methods are written clearly and are easy to follow. My main concern is that the paper does not provide enough background to help the reader understand the results, particularly with regards to the meaning and calculation of the cross term, which is discussed at length in the results section. I suggest an expansion of the introduction section to include: (1) more background on previous non-linearity studies (2) and studies that previously quantified the cross term (if any). Furthermore, an addition to the methods section of: (1) the carbon cycle feedback framework $(\beta, \gamma)$ and (2) the meaning of the cross term and how it is calculated under $CO_2$ and non-$CO_2$ forcing.

A few minor comments are included below:

**Minor comments**

L19: I suggest using the term 'climate-carbon cycle feedback' instead of temperature-driven feedback, since that is the terminology most used in the field.

L20: Is this sentence correct? From my understanding, the $CO_2$ forcing drives both carbon cycle feedbacks through changes in $CO_2$ concentration and temperature, whereas the non-$CO_2$ forcing drives the climate carbon cycle feedback only through changes in temperature. Please clarify.

L38: Acronym 'GHG' not introduced - I suggest writing greenhouse gas in full here.

L50: Please specify which forcing components were included in the Richardson et al. (2019) study. If the study included the response to $CO_2$ and non-$CO_2$ forcing, I suggest briefly discussing the results from this study in your introduction section, and if possible, comparing these results to your results in your discussion section.

L58: This may be a good point to link non-$CO_2$ forcing to the climate-carbon cycle feedback. Non-$CO_2$ forcing induces warming => capacity of the land and ocean sinks reduces => atmospheric $CO_2$ concentration and temperature affected. It may also help to explain why the non-$CO_2$ concentration-carbon feedback is not relevant.

L60: It may help readers to preface this paragraph with a brief description of how the two carbon cycle feedbacks work under increasing and decreasing $CO_2$ concentrations. This will make it easier to understand L62 where you state the results from your Melnikova et al. (2021) study.

L69-71: This sentence is too long. For clarity, please separate the two research questions using (1) and (2) or a semi-colon.

L81: Please clarify which climate factors you are referring to here.

L120: From my understanding of the table format, experiments are above the horizontal line, while combinations of experiments are below the horizontal line. This is why I am surprised that the [CO2bgc+non-CO2] experiment is above the line. Is this an experiment or an addition of two separately run experiments? If it is indeed an experiment, then I assume you prescribed both $CO_2$ forcing and non-$CO_2$ forcings, then specified the piControl $CO_2$ concentration in the radiation code? If so, that would mean that the only warming seen in that experiment would be $CO_2$ physiological warming, so how then can non-$CO_2$ $\gamma$ be included in this experiment? Please clarify.

On the same note, is the additional combination [CO2bgc+non-CO2]-[CO2bgc] necessary? It looks like we could get at non-CO2 $\gamma$ by taking the difference between [CO2+non-CO2] and [CO2] and this would give the cross term as well. Is there a benefit to using [CO2bgc+non-CO2]-[CO2bgc] over [CO2+non-CO2]-[CO2]?

In the 4$^{th}$ column, the first two combinations of experiments seem to be missing the $\Delta U_\gamma$ components.

Figure 1: I would like to commend the authors on this figure – it complements the methods section very nicely.

L146: Section 3.1 assumes that readers have a solid grasp of the carbon cycle feedback framework and the feedback parameters $(\beta, \gamma)$ used, which may not be the case. I suggest prefacing this section with a brief description of carbon cycle feedback parameters (equations for quantification, units and sign convention) before introducing $\Delta U$.

L184: I suggest citing Zickfeld et al. (2011) here.

Figure 2: Is the last column of panels on Figure 2 necessary? I notice that these figures are hardly referenced.

Also, I suggest using a different colour for either the $CO_2$ or $CO_2$bgc lines? The two are compared several times in the text but the colours are difficult to distinguish on the figure panels.

L219: What is the reason for the higher sensitivity to non-$CO_2$ forcing than $CO_2$ forcing?

L262: It appears that the figure in the paper referenced – Chimuka et al. (2023) – shows little hysteresis in autotrophic respiration and GPP, and not in heterotrophic respiration as mentioned in the text.

L283-284: Are there merits to attributing the cross term to $\gamma$ rather than keeping it as a separate term?

---

## Author Comment (AC1)

Our responses are in black, marked as **[Response]**, and the comments of the Reviewers are in purple, marked as **[Comment]**. In our responses, we mark the changes in the manuscript with shading and separate comments using "**********".

**Reviewer #1 (Remarks to the Author):**

The authors investigate carbon cycle feedbacks under CO2 and non-CO2 GHG forcings. Since non-CO2 GHG lead to warming only, the CO2 concentration induced component of the carbon cycle feedbacks is missing for this forcing. This motivates the authors to investigate what has been termed "non-linearity of carbon cycle feedbacks" in previous studies, but with a focus on non-CO2 forcings. The authors use an impressive set of idealized model experiments to separate the different feedback components. The manuscript is generally well written, well structured, and the methods are sound and well described, although some parts of the manuscript could be improved in clarity and I found some of the results difficult to understand (see my comments below). There are only very few studies dealing with the interactions of non-CO2 GHG forcing and the carbon cycle, even though non-CO2 GHG reduction will be an important climate mitigation measure in pathways that limit global warming to below 2 degrees. Although the main results do not seem to be very surprising, I believe this study is a valuable contribution to this field and I would recommend publication in Earth System Dynamics after my comments listed below have been addressed.

We thank Dr. Schwinger for taking the time to read the manuscript and provide detailed and insightful comments and suggestions that helped to improve the manuscript.
* * *
**General comments**

**[Comment 1]**

The topic of this study is complicated and not easy to grasp for a reader without specific knowledge of the carbon-cycle feedback literature. I would therefore encourage the authors to critically review their introduction and provide more explanation of the basic concepts and how they are related to the main topic of the study, the differences between CO2 and non-CO2 GHG forcings. More specifically, I think a link between the non-linearity of carbon cycle feedbacks and the feedbacks due to non-CO2 GHG needs to be made, given that this topic is discussed quite extensively later in the manuscript. It would be a good idea to add **a paragraph to the introduction that deals with the fact (and the causes for) that**

**temperature mediated feedbacks can be different under rising or constant CO2, and**
**that this is the main difference between CO2 and non-CO2 GHG mediated feedbacks.**
Here it would be also pertinent to cite the two (to my knowledge) studies that have investigated
the topic of non-linearity previously (Zickfeld et al. 2011 and Schwinger et al. 2014, both
studies did not deal with non-CO2 forcings). Also, in the Methods and Table 1, there are some
sources of confusion, which should be addressed (see my specific comments below).
  **On a related note, why do the authors not go a step further and introduce a new**
**symbol for the cross term?** A clear definition of the "non-linear" or "cross-term" has been
hampered by the fact that in the first studies using the beta/gamma framework (Friedlingstein
et al. 2003, 2006), gamma was defined by [CO2]-[CO2bgc]. For this reason, also later studies
that actually had a [CO2rad] simulation available continued using the term gamma for both
climate carbon feedbacks [CO2rad] and [CO2]-[CO2bgc], as the authors mention themselves.
This study might be a good opportunity to clean up with this "notational mess"?

**[Response]**

  We are grateful for this comment and for the insight around the "notational mess". As
suggested, in Introduction we added a paragraph that introduces the nonlinearity concept with
the citations on the suggested studies.

  The weakening of land and ocean carbon sinks due to non-$CO_2$ GHGs underscores
the importance of understanding the differences in carbon cycle feedbacks between $CO_2$ and
non-$CO_2$ GHGs. Only the changes in $CO_2$ concentrations are associated with the carbon-
concentration ($\beta$) feedback, that is the response of the land and ocean carbon uptake to the
changes in $CO_2$ concentration, mainly via the stimulation of photosynthesis through $CO_2$
fertilisation effect over land and the solubility pump over the ocean. The changes in both $CO_2$
and non-$CO_2$ concentrations are associated with the carbon-climate feedback ($\gamma$), that is the
response of the land and ocean carbon uptake to climate change, mainly via the increased
plant and soil respiration over land and reduction of the $CO_2$ solubility in the ocean with
warming (Arora et al., 2013; Schwinger et al., 2014; Zickfeld et al., 2011). Under changing
$CO_2$ concentrations, land and ocean carbon storages are simultaneously exposed to the
carbon-concentration and carbon-climate feedbacks. However, the interaction between these
feedbacks can introduce a non-linearity in the system, whereby the combined effect is not
simply the sum of individual feedbacks. Thus, temperature-mediated feedback can differ
under changing versus constant $CO_2$ levels, an important distinction when comparing $CO_2$ and
non-$CO_2$ GHG feedback mechanisms. Here, it is also important to acknowledge that other
factors, such as time lags and potential irreversibilities in the climate system, may also contribute to these differences (Boucher et al., 2012; Chimuka et al., 2023; Schwinger et al., 2014).

Previous studies investigated the nonlinearity in the carbon cycle feedback and revealed that the nonlinearity, or the cross term, may be comparable in size with γ (Schwinger et al., 2014; Zickfeld et al., 2011). They attributed the nonlinearity to the different responses of the land biosphere to the temperature changes, depending on the presence or absence of the $CO_2$ fertilisation effect, as well as the weakening of ocean circulation and mixing between water masses of different temperatures. However, these studies did not consider non-$CO_2$ GHGs.

We also fixed the mix-up in Table 1.

Following the Reviewer's suggestion, we introduced a symbol χ for the cross-term. We modified the Methods section to clarify the reasoning behind the need for the new symbol.

Zickfeld et al. (2011) and Schwinger et al. (2014) demonstrated that Eq. (4) includes the residual term ε, which can be derived from the difference between [$CO_2$] – [$CO_2$bgc] and [$CO_2$rad] - [piControl], using Eq. (5):

$$\varepsilon = \Delta U_{COU} - \Delta U_{BGC} - \Delta U_{RAD}. \qquad (5)$$

These studies revealed that the residual 'nonlinearity' term depends on both $CO_2$ concentration and climate change, and it can be of the same order of magnitude as the γ term. Here, we propose attributing the residual nonlinearity to a cross term, associated with the nonlinearity feedback χ. Although many recent studies continued to attribute χ to the γ feedback—partly due to the absence of the [$CO_2$rad] experiment in some experimental designs, and also because this approach has been widely established in earlier research (Friedlingstein et al., 2003, 2006)—we show that these metrics become less well-defined when examining the effects of both $CO_2$ and non-$CO_2$ GHGs on the carbon cycle.
* * *
**[Comment 2]**

In the section on the physical climate (section 3.1), the strongest warming is found in [CO2rad], but it is not explained why. [CO2rad] is warmer, particularly in the Arctic, than both [CO2] and [nonCO2], if I am not mistaken. Results show no very strong CO2 physiological warming in [CO2bgc], but nevertheless the CO2 physiological warming is used to explain the differences in simulations several times (e.g. lines 221-222), and it remains completely unclear to me why then [CO2rad] is the warmest simulation? **In previous studies, the strongest CO2 physiological warming was found in the Arctic region for CMIP5 ESMs (Park et al. 2020),**

**with significant regional SAT contributions. This study, which includes the predecessor**
**ESM IPSL-CM5A-LR, could be mentioned in the context of the CO2 physiological**
**warming.** In the present study, the authors find the CO2 induced total warming smaller than
the radiative warming alone in high latitudes (line 224, Fig. S5e), which is opposite from the
results of the Park et al. study. This needs at least to be mentioned and if possible some
explanation should be provided (the authors mention differences in snow albedo as an
explanation, but this is rather a consequence than a cause of the different surface
temperatures?).

**[Response]**

Indeed, we completely missed this point in the original manuscript. In the revised
version, we added discussion on the larger warming in the [$CO_2$rad] compared to the [$CO_2$]
experiment. We added some discussion, including a comparison with the findings of Park et
al. (2020). We also revised Fig. S3 to include the [$CO_2$rad] - [$CO_2$] combination.

We tested several potential mechanisms that could lead to a larger warming in
[$CO_2$rad] compared to the [$CO_2$] experiment. Particularly, Park et al. (2020) describe two
contrasting effects of $CO_2$ fertilisation: (1) $CO_2$ leads to reduction in the stomatal conductance,
which in its turn decreases evaporative cooling, and (2) $CO_2$ leads to higher leaf area index,
which (i) increases evaporative cooling and (ii) decreases albedo, which also leads to warming.
We cannot approve either of these mechanisms because land evaporation is slightly higher in
the [$CO_2$rad] experiment (Fig S5).

[Figure]

**Figure S5. (a–d) Global, (e–h) land and (i–l) ocean annually-averaged changes in evapotranspiration (mm year$^{-1}$) as a function of (a, e, i) time (year), (b, f, j) $CO_2$ concentration (ppm) / $CH_4$ concentration (ppb, only for [nonCO$_2$]), (c, g, k) GSAT ($^{\circ}$C) and (d, h, l) cumulative GSAT ($^{\circ}$C).**

The behaviour of the IPSL-CM6A-LR model remains the same in other similar experiments. Comparison of CMIP6 1pctCO2 (fully-coupled experiment with 1% $CO_2$ increase per year) and 1pctCO2-rad (same but radiatively-coupled) shows that, in agreement with our results, the fully coupled experiment produces a slightly higher surface air temperature increase, especially in the northern high latitudes, at moderate $CO_2$ levels (Figure R1). Similar behaviour can be seen in the GFDL-ESM4 simulations but is absent in the NorESM2-LM. As noted by the Reviewer, the ensemble size in our study is small and the effects of the model's internal variability should be considerable.

[Figure]

**Figure R1. Time series of (left) northern high-latitude (>60⁰ N) and (right) global surface air temperature increase (K) in the radiatively-, biogeochemically- and fully-coupled 1pctCO2 experiments by selected CMIP6 ESMs. The vertical dotted line indicates year, when the experiment's CO₂ concentration is nearly equal to the maximum CO₂ concentration (403 ppm) of this study.**

We have added the following discussion of the differences between our results and those of Park et al. (2020).

The combined effects of $CO_2$ physiological and radiative forcing do not lead to more warming, as the radiative forcing alone ([$CO_2$rad] experiment) leads to a slightly higher global temperature increase compared to the coupled [$CO_2$] experiment (Fig. 2a, b). This temperature difference is particularly evident in the Arctic region (Fig. S3a). Our findings differ from those of a CMIP5 intercomparison study, which reported that $CO_2$ physiological warming amplifies the Arctic warming (Park et al., 2020). The study showed that the $CO_2$ physiological effect contributes to high-latitude warming by reducing evaporative cooling due to stomatal closure under elevated $CO_2$ levels. In contrast, we observe higher evapotranspiration in the [$CO_2$rad] compared to the [$CO_2$] experiment (Fig. S5), which is probably a consequence of the lower warming in the [$CO_2$] experiment. In our study, the greater warming in the [$CO_2$rad] experiment may be driven by increased surface albedo, especially over the Arctic Ocean (Fig. S3b). While the underlying causes remain unclear, this pattern appears consistent in other experiments conducted with IPSL-CM6A-LR under moderate $CO_2$ levels (not shown). Because the ensemble size in our study is limited and the effects of the model's internal variability should be considerable, future research should validate the robustness of our findings with larger ensemble simulations.
* * *
**[Comment 3]**

Table 1 is somewhat confusing. Column 4 refers only to beta and gamma such that both experiments [CO2rad] and [CO2]-[CO2bgc] appear to be the same (they include the carbon cycle feedback "CO2 gamma"), but it is not mentioned that the cross-term is present in [CO2]-[CO2bgc]. The same is true for [nonCO2] and [CO2bgc+nonCO2]-[CO2bgc]. Also, in the 5[th] column the only term listed for [CO2]-[CO2bgc] is the cross term, while the actual gamma-term is missing. Again, the same is true for [CO2bgc+nonCO2]-[CO2bgc]. In the footnotes, the terms $\Delta U_{\gamma,CO2physiological}$ and $\Delta U_{\beta\gamma,CO2physilogical}$ are not defined anywhere. I would suggest to just say that the warming from the physiological CO2 forcing is assumed to be negligible.

**[Response]**

Columns 4 ("Included carbon cycle feedback") and 5 ("Included carbon cycle terms from Eq.2") in the original manuscript included such information, and apparently column 4

adds more confusion than clarity to the description of the experimental design. Thus, in the
revised manuscript we delete it.

As the Reviewer pointed out, the terms in column 5 had errors on the included terms,
which is now corrected. We also remove the original explanation on the $\Delta U_{\gamma,CO2physiological}$ and
$\Delta U_{\beta\gamma,CO2physilogical}$ terms and added the following instead:

*according to equations by Etminan et al. (2016), warming from the physiological $CO_2$
forcing is assumed to be negligible.

**********

**[Comment 4]**

In the abstract (line 21-22), even if Arora et al 2020 and Schwinger et al. 2014, did not
use the term "cross-term" but "non-linearity", the results are consistent with these studies. So
I would suggest adding "consistent with previous studies that considered CO2 forcing only".
**[Response]**

We agree with the suggestion and revised the abstract accordingly.

We introduce a framework, consistent with previous studies that focused exclusively
on $CO_2$ forcing, to separate the carbon-climate feedback into the temperature and cross terms.
Our findings reveal that these feedback terms are comparable in magnitude for the global
ocean. This underscores the importance of considering both terms in carbon cycle feedback
framework and climate change mitigation strategies.

**********

**Specific comments**

**[Comment 1]**

Equation 2: It might be pertinent to cite Schwinger et al. 2014 here, who used the
Taylor expansion to define "nonlinearity" of carbon cycle feedbacks. Please double check the
factor 1/2 in the cross-term (also in Equation 5), which is wrong I believe (only the quadratic
terms have the factor of 1/2).
**[Response]**

As suggested, we added the citation, changing text to:

Following Schwinger et al. (2014) the formulation can be expanded to a Taylor series…

Besides we agree that the factor of 1/2 is wrong here, removed.

**********

**[Comment 2]**

Equations 3-4: Why are the quadratic terms included here? They cannot be quantified,
so they belong to the residual term in the context of this study.

**[Response]**

We respectfully disagree, because via our analysis, we conclude that second-order
terms (quadratic terms) cannot be neglected. We think it is necessary to show them
consistently with the quadratic term that is needed to define the cross term ($\Delta U_\chi$), as shown
below.

$$\Delta U_\beta = \frac{\partial U}{\partial C_{CO2}}\Delta C_{CO2} + \frac{1}{2}\frac{\partial^2 U}{\partial C_{CO2}{}^2}(\Delta C_{co2})^2 + Res., \tag{7}$$

$$\Delta U_\gamma = \frac{\partial U}{\partial T}\Delta T + \frac{1}{2}\frac{\partial^2 U}{\partial T^2}\Delta T^2 + Res., \tag{8}$$

$$\Delta U_\chi = \frac{\partial^2 U}{\partial C_{CO2}\,\partial T}\Delta T\Delta C_{CO2} + Res.. \tag{9}$$

**********

**[Comment 3]**

Line 19-20: Please double check the sentence: Shouldn't this be the other way around
– "Non-CO2 forcing primarily affects temperature driven feedbacks…" or did I misunderstand
something here?

**[Response]**

This indeed should be the other way around, now corrected.

$CO_2$ forcing affects both carbon-climate and carbon-concentration feedbacks, whereas non-
$CO_2$ gases influence only the carbon-climate feedback.

**********

**[Comment 4]**

Line 22: It is a bit unclear what "both components" refers to. Also, non-CO2 forcing are
usually considered in Earth system modelling, e.g., in SSP scenarios. Please reword this
sentence to make the main conclusion of this paper clearer.

**[Response]**

We changed the wording to "feedback terms" to be consistent with the previous sentence. We further reworded the last sentence of the abstract by rewording "considered in Earth system modelling" to "considered in carbon cycle feedback framework". Now it reads as follows:

Our findings reveal that these feedback terms are comparable in magnitude for the global ocean. This underscores the importance of considering both terms in carbon cycle feedback framework and climate change mitigation strategies.
* * *
**[Comment 5]**

Line 75-81: "like many contemporary models" could be made more specific by saying "like all other ESMs participating in CMIP6" or similar. Generally, I think this paragraph is not necessary here. These are idealized concentration-driven experiments, so why discuss the lack of CH4 and N2O emission-driven capability in the Introduction? Particularly, since a section on "limitations" exists at the end of the manuscript. I would suggest deleting this paragraph and move parts of the text to Section 4.

**[Response]**

We followed the Reviewer's suggestion by deleting this paragraph and moving part of it to the section on study limitations as follows:

However for this study, the use of the model is justified because current changes in $CH_4$ and $N_2O$ concentrations are primarily driven by anthropogenic sources, suggesting that the absence of interactive modules of natural sink/source processes does not significantly affect the representation of natural variability trends for the $CH_4$ and $N_2O$ concentration (Nakazawa, 2020; Palazzo Corner et al., 2023; Zhu et al., 2013).
* * *
**[Comment 6]**

Line 210-214: Although the physiological warming might be "significant" it is still quite small. Also, I would suggest being more careful here (and elsewhere in the manuscript), since the ensemble size is small and decadal scale variability can still be present in the ensemble mean. For example, the "significant CO2 physiological warming" in [CO2bgc] over "the high latitudes of land and ocean during stabilization period" could very well be an effect of AMOC, which happens to be significantly stronger over much of the stabilization period of [CO2bgc]

compared to [piControl] in two of three ensemble members (Fig. S1a).

**[Response]**

We agree and, thus, changed the paragraph to include a more careful statement.

The $CO_2$ physiological warming that can be quantified by comparing [$CO_2$bgc] with

[piControl] is small (green line in Fig. 2). Spatially, some differences are ubiquitous over land, e.g., $CO_2$ physiological warming persists over Eurasia during the ramp-up period, and over the high latitudes of both land and ocean during the stabilisation period (Fig. S3a). A larger ensemble size of model simulations would be required to investigate these differences more thoroughly. In our following analysis on carbon cycle feedbacks, we assume the $CO_2$

physiological warming to be negligible.

**********

**[Comment 7]**

Line 219: "… the higher sensitivity to non-CO2 forcing compared to CO2 forcing". This should be the other way round (SAT is higher under CO2 forcing)?

**[Response]**

Indeed, this should be the other way round, corrected.

**********

**[Comment 8]**

Line 220-221: "The combined effect of CO2 physiological and radiative forcing leads to more warming in the coupled [CO2] experiment compared to both the [CO2rad] experiment."

I guess the "both" should be deleted? Also, I cannot see this in Fig 2a, here [CO2rad] shows a stronger warming than [CO2].  This is consistent with the figures in the supplementary, which also show that [CO2rad] seems to be warmer than both [CO2] and [nonCO2], particularly in the high latitudes (Fig S3a). What is the reason for this? Also, as mentioned above, this is different from the CMIP5 study of Park et al. 2020.

**[Response]**

We deleted the unnecessary "both" term. We agree with the comment and revised the
paragraph, as described in our response to General comment 2.

**********

**[Comment 9]**

Line 223-224: "…the CO2-induced total surface warming is larger than CO2-induced
radiative warming almost everywhere, except for the high northern latitudes over the land and
ocean (Fig. S3)." I can't see this from Fig S3, because [CO2]-[CO2rad] is not shown there.
Again, the most striking difference is that [CO2rad] is warmer than [nonCO2], particularly in
high latitudes (and by comparison with the next column also warmer than [CO2] in the high
latitudes. What is the reason for this difference?

**[Response]**

We added the [CO2]-[CO2rad] experiment to Fig. S3 in the revised manuscript version.
We also revised the discussion, as described in our response to General comment 2.

[Figure]

**Figure S3. Spatial variation of three-member-ensemble mean changes in (a) surface**
**temperature (ºC) and (b) surface albedo averaged over 20 years at the end of (first rows)**

**ramp-up, (middle rows) ramp-down, and (bottom rows) stabilisation phases relative to**
**piControl under selected scenarios. We draw only grids significantly different from**
**piControl ($p < 0.1$ based on t test, N=60) and between [CO$_2$], [CO$_2$rad] and [nonCO$_2$]**
**experiments using three ensemble members ($p < 0.1$ based on t test, N=60).**

**********

**[Comment 10]**

Line 238-243: This paragraph is very confusing. It seems to repeat things that have
been explained in the Methods section, but in a way that I doubt is helpful for the reader. I
would suggest either rewording and expanding this paragraph or deleting it. Again, the terms
$\Delta U_{\gamma,CO2physiological}$ and $\Delta U_{\beta\gamma,CO2physilogical}$ have never been defined in the manuscript.
**[Response]**

We now deleted the paragraph, as in section 3.1 we state that we assume CO$_2$
physiological warming to be negligible.

**********

**[Comment 11]**

Table 2: While CO2 (and non-CO2 GHG) concentrations are all the same in the
different concentration driven experiments, this is not the case for the temperature increase.
For example, SAT is 10-15% lower for [nonCO2] compared to [CO2] and [CO2rad] (estimated
from Fig.2). Therefore, I am wondering if it would not make more sense to give values for
gammas in this table? I would expect $\Delta U_{\gamma,nonCO2}$ be somewhat lower than $\Delta U_{\gamma,CO2rad}$ just
because of the lower temperature increase, while it is actually gamma which makes the most
useful comparison between the simulations. More importantly, how are the cross-term carbon
uptakes (first line in the lower part of the table) calculated? Shouldn't this be the difference
between the second and fourth line of the upper part of the table? I cannot see this is the case.
**[Response]**

We chose to report values of cumulative fluxes rather than those of the feedback
parameter because we wanted to show the changes for ramp-up, ramp-down and stabilisation
periods. Estimation of the feedback parameter values for the end of the ramp-up period is
possible and we included it in the newly added Table S1. We also added a column with
experiment's peak temperatures (mean temperature at the end of ramp-up period) to Table 2.

However, due to the lagged responses of both temperature and carbon fluxes,
estimation of feedback parameters for the ramp-down period is more challenging. Furthermore, we would face numerical issues for calculating carbon cycle feedback parameters for stabilisation and total periods.

The newly added Table S1 (below) shows larger negative γ for land and smaller negative γ for ocean in the [$CO_2$rad] compared to [non$CO_2$] experiment, but these differences are not statistically significant.

We confirmed some errors in the table for the means of the cross terms. We corrected them in the revised manuscript.

**Table 2. Cumulative $CO_2$ and climate change-driven changes in the land and ocean carbon fluxes (GtC), shown as three- member ensemble mean. The ± indicates one standard deviation among the three members. Note that all experiments are analysed relative to their [piControl] counterparts.**

| Experiment | Max. warming (K)* | Terms | Years 1-50 (ramp-up) Land | Ocean | Years 51-100 (ramp-down) Land | Ocean | Years 101-150 (stabilisation) Land | Ocean | Total Land | Ocean |
|---|---|---|---|---|---|---|---|---|---|---|
| [$CO_2$bgc] | 0.1±0.0 | $\Delta U_{\beta,CO2}$ | 179.3±2.2 | 103.8±0.7 | 16.3±6.0 | 19.7±1.0 | 106.3±0.4 | 32.1±0.8 | 59.4±1.8 | 53.4±0.9 |
| [$CO_2$rad] | 1.1±0.1 | $\Delta U_{\gamma,CO2}$ | -18.6±2.4 | -2.4±0.2 | 4.5±5.7 | 0.0±1.4 | -11.4±4.0 | 0.3±1.9 | -2.2±2.0 | -2.1±0.6 |
| [non$CO_2$] | 0.9±0.1 | $\Delta U_{\gamma,nonCO2}$ | -14.8±3.6 | -2.3±1.0 | 1.3±6.0 | -0.5±1.3 | -10.2±2.4 | 1.3±0.5 | -2.5±2.9 | -1.6±0.2 |
| [$CO_2$] − [$CO_2$bgc] | 1.0±0.0 | $\Delta U_{\gamma,CO2} + \Delta U_{\chi,CO2}$ | -14.7±1.0 | -4.5±0.2 | 3.8±3.0 | 1.4±1.0 | 8.2±1.9 | 0.6±0.5 | -3.8±3.1 | 5.5±1.4 |
| | | $\Delta U_{\chi,CO2}$ | 3.9±2.1 | 2.2±0.4 | 0.7±7.3 | 1.4±0.5 | 3.2±5.9 | 0.2±2.3 | 1.6±5.0 | 3.4±2.0 |
| | | $\Delta U_{\chi,nonCO2}$ | 3.5±1.5 | 1.6±1.0 | 2.8±7.9 | 0.1±1.1 | 1.6±1.5 | 1.0±1.3 | 2.0±4.0 | 2.5±1.0 |
| | | $\Delta U_{\gamma,CO2} - \Delta U_{\gamma,nonCO2}$ | 3.8±5.0 | 0.1±0.9 | 3.2±3.6 | 0.5±0.9 | 1.2±2.6 | 1.0±2.2 | 0.3±2.7 | 0.5±0.7 |
| | | $\Delta U_{\chi,CO2} - \Delta U_{\chi,nonCO2}$ | 0.4±3.5 | 0.6±0.7 | 3.5±3.4 | 1.6±1.0 | 1.6±7.4 | 1.3±3.3 | 3.6±5.5 | 1.0±2.8 |

* defined as the mean ΔGSAT during years 41-60.

**Table S1. Changes in the carbon cycle feedback parameters for land and ocean at the end of the ramp-up period, shown as three- member ensemble mean. The ± indicates one standard deviation among the three members. We use temperature of the fully coupled experiments to estimate γ and χ feedbacks.**

| Experiment | Terms | Years 1-50 (ramp-up) Land | Ocean |
|---|---|---|---|
| [$CO_2$bgc] | $\beta_{CO2}$ (GtC ppm$^{-1}$) | 1.51 ± 0.02 | 0.88 ± 0.01 |
| [$CO_2$rad] | $\gamma_{CO2}$ (GtC K$^{-1}$) | -17.02 ± 1.44 | -2.17 ± 0.14 |
| [non$CO_2$] | $\gamma_{nonCO2}$ (GtC K$^{-1}$) | -16.74 ± 4.12 | -2.58 ± 1.19 |
| [$CO_2$] - [$CO_2$bgc] - [$CO_2$rad] | $\chi_{CO2}$ (GtC ppm$^{-1}$ K$^{-1}$) | 0.03 ± 0.02 | -0.02 ± 0.0 |
| [non$CO_2$bgc] - [$CO_2$bgc] - [non$CO_2$] | $\chi_{nonCO2}$ (GtC ppm$^{-1}$ K$^{-1}$) | 0.03 ± 0.01 | -0.01 ± 0.01 |
* * *
 **[Comment 12]**

    Line 282-284: As mentioned above, it is a choice to "attribute" the cross-term to the

 carbon-climate feedback, which makes sense in the context of previous studies. But I don't

 see why this would be necessary, and I would encourage the authors to drop this attribution

 and just go ahead with beta, gamma, and the cross-term (as mentioned above, maybe

 introduce a new symbol for the cross term?).

 **[Response]**

    We thank the Reviewer for the encouragement. We have divided the original "Carbon-

 Climate Feedback" section by creating a new section titled "Nonlinearity in Carbon Cycle

 Feedback." Additionally, we introduce the symbol χ to represent the cross term.
* * *
 **[Comment 13]**

    Line 304: "larger climate change driven carbon source" is not precise. It is rather a

 larger climate change driven reduction of the ocean sink. The ocean remains a sink throughout.

 Same comment applies for line 312.

 **[Response]**

    Revised accordingly.

    Over ocean, the contribution from the χ term leads to a greater reduction in the carbon

 sink driven by climate change (Fig. 3).

    ..

    Spatially, while the Southern Ocean remains the largest ocean carbon sink in all

 considered experiments involving atmospheric $CO_2$ changes, it, along with the Atlantic Ocean,

 undergoes the largest climate change-driven reduction in carbon sink (Fig. 4).
* * *
 **[Comment 14]**

    Line 324: Why would reducing non-CO2 GHG only change $\Delta U_\gamma$? By changing

 temperature, the cross-term would be affected, too.

 **[Response]**

Agreed, changed to "implies alteration of $\Delta U_\gamma$ and $\Delta U_\chi$ terms."

**********

**[Comment 15]**

Line 369-370: Again, the highest GSAT is found in [CO2rad] which is inconsistent with
this conclusion.

**[Response]**

We removed this sentence from the Conclusions in the revised manuscript.

**Technical comments**

**[Comment 1]**

Line 37: delete "over"

**[Response]**

Deleted

**********

**[Comment 2]**

Line 69: consider changing to "to clarify whether the climate responses to declining
CO2 and non-CO2 GHGs differ globally and regionally."

**[Response]**

Changed to the suggested formulation.

**********

**[Comment 3]**

Line 86: Place reference to Boucher et al. 2020 after the model name, not after CMIP.
Replace CMIP by CMIP6

**[Response]**

Changed accordingly.

**********

**[Comment 4]**

Line 96: Confusing sentence, please consider rewording. Maybe "… between a model experiment with perturbed GHG concentration but fixed sea surface and ice temperatures and a control simulation with pre-industrial GHG concentrations." or similar.

**[Response]**

Changed to the suggested formulation.

**********

**[Comment 5]**

Line 108: "referred to" could be understood as if the effective concentrations are used in the text and figures. I would suggest rewording this sentence.

**[Response]**

We reworded the second half of the sentence, which now reads:

The effective concentrations of $CH_4$ and $N_2O$ are used as input to the radiative transfer scheme of the climate model throughout the rest of this study. In the text and figures, these are presented as the actual (equivalent) concentrations.

**********

**[Comment 6]**

Line 156: Delete "atmospheric CO2 induced".

**[Response]**

Deleted.

**********

**[Comment 7]**

Line 199: thermostatic -> thermosteric

**[Response]**

Corrected.

**********

**[Comment 8]**

Line 201: Consider replacing "under considered timescale" by "within the time-horizon considered here" or similar.

**[Response]**

Changed as suggested.

**********

**[Comment 9]**

Line 250: "… which induces carbon sink…" -> "which represents the CO2 induced carbon sink…"

**[Response]**

Changed as suggested.

**********

**[Comment 10]**

Line 254: Complicated sentence. Why not say "Over the ocean beta is positive (carbon sink) in all regions …"

**[Response]**

Changed, as suggested. The sentence now reads:

Over the ocean $\beta$ is positive (carbon sink) in all regions during the ramp-up period (Fig.

4).

**********

**[Comment 11]**

Line 278: What do you mean by "prolonged duration of beta"? Please clarify.

**[Response]**

Changed to "the extended period of large β influence".

**********

**[Comment 12]**

Line 286: Please spell out what "equivalent" means (within one standard deviation?).

**[Response]**

Added (within one standard deviation uncertainty range)".

**********

**[Comment 13]**

Line 287: Remove subscript betas before "in Table 2".

**[Response]**

Corrected

**********

**[Comment 14]**

Line 295: the gamma -> gamma

**[Response]**

Corrected.

**********

**Reviewer references**

Park, SW., Kim, JS. & Kug, JS. The intensification of Arctic warming as a result of $CO_2$

physiological forcing. *Nat Commun* **11**, 2098 (2020). https://doi.org/10.1038/s41467-020-

15924-3

Schwinger, J., and Coauthors, 2014: Nonlinearity of Ocean Carbon Cycle Feedbacks in CMIP5 Earth System Models. *J. Climate*, **27**, 3869–3888, https://doi.org/10.1175/JCLI-D-

13-00452.1.

Zickfeld, K., M. Eby, H. D. Matthews, A. Schmittner, and A. J. Weaver, 2011:

Nonlinearity of Carbon Cycle Feedbacks. *J. Climate*, **24**, 4255–4275, https://doi.org/10.1175/2011JCLI3898.1.

**References**

**References**

Arora, V. K., Boer, G. J., Friedlingstein, P., Eby, M., Jones, C. D., Christian, J. R., Bonan, G.,
Bopp, L., Brovkin, V., Cadule, P., Hajima, T., Ilyina, T., Lindsay, K., Tjiputra, J. F., and Wu, T.:
Carbon–Concentration and Carbon–Climate Feedbacks in CMIP5 Earth System Models, J.
Climate, 26, 5289–5314, https://doi.org/10.1175/JCLI-D-12-00494.1, 2013.

Boucher, O., Halloran, P. R., Burke, E. J., Doutriaux-Boucher, M., Jones, C. D., Lowe, J.,
Ringer, M. A., Robertson, E., and Wu, P.: Reversibility in an Earth System model in response
to CO 2 concentration changes, Environ. Res. Lett., 7, 024013, https://doi.org/10.1088/1748-
9326/7/2/024013, 2012.

Chimuka, V. R., Nzotungicimpaye, C.-M., and Zickfeld, K.: Quantifying land carbon cycle
feedbacks under negative CO2 emissions, Biogeosciences, 20, 2283–2299,
https://doi.org/10.5194/bg-20-2283-2023, 2023.

Etminan, M., Myhre, G., Highwood, E. J., and Shine, K. P.: Radiative forcing of carbon dioxide,
methane, and nitrous oxide: A significant revision of the methane radiative forcing,
Geophysical Research Letters, 43, 12,614-12,623, https://doi.org/10.1002/2016GL071930,
2016.

Friedlingstein, P., Dufresne, J.-L., Cox, P. M., and Rayner, P.: How positive is the feedback
between climate change and the carbon cycle?, Tellus B: Chemical and Physical Meteorology,
https://doi.org/10.3402/tellusb.v55i2.16765, 2003.

Friedlingstein, P., Cox, P., Betts, R., Bopp, L., von Bloh, W., Brovkin, V., Cadule, P., Doney,
S., Eby, M., Fung, I., Bala, G., John, J., Jones, C., Joos, F., Kato, T., Kawamiya, M., Knorr,
W., Lindsay, K., Matthews, H. D., Raddatz, T., Rayner, P., Reick, C., Roeckner, E., Schnitzler,
K.-G., Schnur, R., Strassmann, K., Weaver, A. J., Yoshikawa, C., and Zeng, N.: Climate–
Carbon Cycle Feedback Analysis: Results from the C4MIP Model Intercomparison, J. Climate,
19, 3337–3353, https://doi.org/10.1175/JCLI3800.1, 2006.

Nakazawa, T.: Current understanding of the global cycling of carbon dioxide, methane, and
nitrous oxide, Proceedings of the Japan Academy, Series B, 96, 394–419,
https://doi.org/10.2183/pjab.96.030, 2020.

Palazzo Corner, S., Siegert, M., Ceppi, P., Fox-Kemper, B., Frölicher, T. L., Gallego-Sala, A.,
Haigh, J., Hegerl, G. C., Jones, C. D., Knutti, R., Koven, C. D., MacDougall, A. H.,
Meinshausen, M., Nicholls, Z., Sallée, J. B., Sanderson, B. M., Séférian, R., Turetsky, M.,
Williams, R. G., Zaehle, S., and Rogelj, J.: The Zero Emissions Commitment and climate
stabilization, Frontiers in Science, 1, https://doi.org/10.3389/fsci.2023.1170744, 2023.

Park, S.-W., Kim, J.-S., and Kug, J.-S.: The intensification of Arctic warming as a result of CO2
physiological forcing, Nature Communications, 11, 2098, https://doi.org/10.1038/s41467-020-
15924-3, 2020.

Piao, S., Wang, X., Park, T., Chen, C., Lian, X., He, Y., Bjerke, J. W., Chen, A., Ciais, P.,
Tømmervik, H., Nemani, R. R., and Myneni, R. B.: Characteristics, drivers and feedbacks of
global greening, Nature Reviews Earth & Environment, 1, 14–27,
https://doi.org/10.1038/s43017-019-0001-x, 2020.

Schwinger, J., Tjiputra, J. F., Heinze, C., Bopp, L., Christian, J. R., Gehlen, M., Ilyina, T., Jones,
C. D., Salas-Mélia, D., Segschneider, J., Séférian, R., and Totterdell, I.: Nonlinearity of Ocean
Carbon Cycle Feedbacks in CMIP5 Earth System Models, Journal of Climate, 27, 3869–3888,
https://doi.org/10.1175/JCLI-D-13-00452.1, 2014.

Zhu, X., Zhuang, Q., Gao, X., Sokolov, A., and Schlosser, C. A.: Pan-Arctic land–atmospheric
fluxes of methane and carbon dioxide in response to climate change over the 21st century,
Environmental Research Letters, 8, 045003, https://doi.org/10.1088/1748-9326/8/4/045003,
2013.

Zickfeld, K., Eby, M., Matthews, H. D., Schmittner, A., and Weaver, A. J.: Nonlinearity of
Carbon Cycle Feedbacks, Journal of Climate, 24, 4255–4275,
https://doi.org/10.1175/2011JCLI3898.1, 2011.

---

## Author Comment (AC2)

Our responses are in black, marked as **[Response]**, and the comments of the Reviewers are
in purple, marked as **[Comment]**. In our responses, we mark the changes in the manuscript
with shading and separate comments using "***********".

**Reviewer #2 (Remarks to the Author):**

This study compares the climate and carbon cycle response to equivalent CO2 and
non-CO2 forcings using a set of idealized concentration-driven simulations. The authors find
that the climate-carbon feedback is dominant under non-CO2 forcing whereas both the
carbon-concentration and climate-carbon feedbacks are important under the CO2 forcing.
Under both CO2 and non-CO2 forcings, the land and ocean carbon uptake due to both
feedbacks is quantified along with a cross term i.e., a term that quantifies the response to
climate change in the presence of CO2 concentration.

The manuscript reads well – the introduction and methods are written clearly and are
easy to follow. My main concern is that the paper does not provide enough background to help
the reader understand the results, particularly with regards to the meaning and calculation of
the cross term, which is discussed at length in the results section. I suggest an expansion of
the introduction section to include: (1) more background on previous non-linearity studies (2)
and studies that previously quantified the cross term (if any). Furthermore, an addition to the
methods section of: (1) the carbon cycle feedback framework ($\beta$, $\gamma$) and (2) the meaning of
the cross term and how it is calculated under CO2 and non-CO2 forcing.

**[Response]**

We thank the Reviewer for taking the time to read the manuscript and provide detailed
and insightful comments that helped to improve the manuscript.

In response to the concern of the Reviewer as to the lack of background of the study,
we expanded the Introduction to have a paragraph on the feedback nonlinearity and existing
studies that investigate and quantify nonlinearity.

The weakening of land and ocean carbon sinks due to non-$CO_2$ GHGs underscores
the importance of understanding the differences in carbon cycle feedbacks between $CO_2$ and
non-$CO_2$ GHGs. Only the changes in $CO_2$ concentrations are associated with the carbon-
concentration ($\beta$) feedback, that is the response of the land and ocean carbon uptake to the
changes in $CO_2$ concentration, mainly via the stimulation of photosynthesis through $CO_2$
fertilisation effect over land and the solubility pump over the ocean. The changes in both $CO_2$
and non-$CO_2$ concentrations are associated with the carbon-climate feedback ($\gamma$), that is the
response of the land and ocean carbon uptake to climate change, mainly via the increased plant and soil respiration over land and reduction of the $CO_2$ solubility in the ocean with warming (Arora et al., 2013; Schwinger et al., 2014; Zickfeld et al., 2011). Under changing $CO_2$ concentrations, land and ocean carbon storages are simultaneously exposed to the carbon-concentration and carbon-climate feedbacks. However, the interaction between these feedbacks can introduce a non-linearity in the system, whereby the combined effect is not simply the sum of individual feedbacks. Thus, temperature-mediated feedback can differ under changing versus constant $CO_2$ levels, an important distinction when comparing $CO_2$ and non-$CO_2$ GHG feedback mechanisms. Here, it is also important to acknowledge that other factors, such as time lags and potential irreversibilities in the climate system, may also contribute to these differences (Boucher et al., 2012; Chimuka et al., 2023; Schwinger et al., 2014).

Previous studies investigated the nonlinearity in the carbon cycle feedback and revealed that the nonlinearity, or the cross term, may be comparable in size with γ (Schwinger et al., 2014; Zickfeld et al., 2011). They attributed the nonlinearity to the different responses of the land biosphere to the temperature changes, depending on the presence or absence of the $CO_2$ fertilisation effect, as well as the weakening of ocean circulation and mixing between water masses of different temperatures. However, these studies did not consider non-$CO_2$ GHGs.

We also expanded the subsection "2.3 Carbon cycle feedback attribution" of the Methods to have more detailed and clear information about the carbon cycle feedback framework and the cross term.

Traditionally, carbon cycle feedback analysis relies on [$CO_2$], [$CO_2$bgc] and [$CO_2$rad] simulations (Arora et al., 2013, 2020; Friedlingstein et al., 2006; Gregory et al., 2009; Schwinger et al., 2014; Schwinger and Tjiputra, 2018; Williams et al., 2019; Zickfeld et al., 2011). The carbon uptake (ΔU) can then be estimated using the well-established carbon cycle feedback framework as a sum of carbon-concentration β parameter (GtC ppm$^{-1}$) multiplied by the changes in the atmospheric $CO_2$ concentration $\Delta C_{CO2}$ (ppm) and carbon-climate γ feedback parameter (GtC K$^{-1}$) multiplied by the changes in surface temperature ΔT (K), using Eq. (1):

$$\Delta U = \beta \times \Delta C_{CO2} + \gamma \times \Delta T + \varepsilon . \tag{1}$$

Here, term ε refers to a residual term.

The β parameter can be estimated from the [$CO_2$bgc] - [piControl], using Eq. (2):

$$\beta = \frac{\Delta U_{BGC}}{\Delta C_{CO2}}, \tag{2}$$

where $\Delta U_{BGC}$ is the carbon uptake in the biogeochemically-coupled experiment [$CO_2$bgc]. The β feedback is associated with strengthening of the land and ocean carbon sink (positive to the land and ocean). Thus, it acts as negative climate feedback (decreasing $CO_2$

content and dampening climate change).

Existing studies derived the γ feedback from the [$CO_2$rad] - [piControl] combination of experiments, using Eq. (3), as well as from the difference between the [$CO_2$] and [$CO_2$bgc]

experiments, hereafter referred to as [$CO_2$] – [$CO_2$bgc], using Eq. (4) (Arora et al., 2013, 2020;

Asaadi et al., 2024; Friedlingstein et al., 2003, 2006; Melnikova et al., 2021):

$$\gamma = \frac{\Delta U_{RAD}}{\Delta T}, \tag{3}$$

$$\gamma = \frac{\Delta U_{COU-BGC}}{\Delta T}, \tag{4}$$

where $\Delta U_{RAD}$ and $\Delta U_{COU\text{-}BGC}$ are the carbon uptake in the radiatively-coupled experiment [$CO_2$rad] and the difference between the COU and BGC experiments [$CO_2$] –

[$CO_2$bgc]. The γ feedback is associated with a weakening of the land and ocean carbon sinks globally, albeit with regional variability (negative to the land and ocean). Thus, it acts as positive climate feedback (increasing $CO_2$ content and accelerating climate change).

**********

**Minor comments**

A few minor comments are included below:

**[Comment 1]**

L19: I suggest using the term 'climate-carbon cycle feedback' instead of temperature- driven feedback, since that is the terminology most used in the field.

**[Response]**

We agree and changed the term to "carbon-climate feedback" to be consistent with existing studies (e.g., Arora et al., 2013; Schwinger et al., 2014).

**********

**[Comment 2]**

L20: Is this sentence correct? From my understanding, the CO2 forcing drives both carbon cycle feedbacks through changes in CO2 concentration and temperature, whereas the non-CO2 forcing drives the climate carbon cycle feedback only through changes in temperature. Please clarify.

**[Response]**

This indeed was erroneous (should be the other way round), now corrected.

$CO_2$ forcing affects both carbon-climate and carbon-concentration feedbacks, whereas
non-$CO_2$ gases influence only the carbon-climate feedback.

**********

**[Comment 4]**
L38: Acronym 'GHG' not introduced - I suggest writing greenhouse gas in full here.
**[Response]**

Added.

**********

**[Comment 5]**
L50: Please specify which forcing components were included in the Richardson et al.
(2019) study. If the study included the response to CO2 and non-CO2 forcing, I suggest briefly
discussing the results from this study in your introduction section, and if possible, comparing
these results to your results in your discussion section.
**[Response]**

We add clarification, now the text reads as follows.

Richardson et al. (2019) revealed spatial and temporal differences in the surface
temperature response to different forcings, such as $CO_2$ and $CH_4$, in part due to the
physiological $CO_2$ warming over the densely vegetated regions that is absent under non-$CO_2$
forcing.

Our findings are consistent with Richardson et al. (2019), which we briefly
acknowledge in the revised manuscript.

When comparing $CO_2$- and non-$CO_2$-induced forcing ([$CO_2$] and [$nonCO_2$]
experiments) at a global scale, our results are consistent with Richardson et al. (2019) who
show the higher surface temperature response of $CO_2$ when compared to $CH_4$.

**********

**[Comment 6]**

   L58: This may be a good point to link non-CO2 forcing to the climate-carbon cycle feedback.

   Non-CO2 forcing induces warming => capacity of the land and ocean sinks reduces => atmospheric CO2 concentration and temperature affected. It may also help to explain why the non-CO2 concentration-carbon feedback is not relevant.

**[Response]**

   We are grateful for this suggestion. Following the Reviewer's comment, we added a linkage of non-$CO_2$ forcing to the climate-carbon cycle feedback to the Introduction as described in our response to the main comment.
* * *
**[Comment 7]**

   L60: It may help readers to preface this paragraph with a brief description of how the two carbon cycle feedbacks work under increasing and decreasing CO2 concentrations. This will make it easier to understand L62 where you state the results from your Melnikova et al. (2021) study.

**[Response]**

   We agree and added a brief description as follows.

   Previous studies have also examined the impact of declining atmospheric $CO_2$ concentrations on the climate and carbon cycle (Boucher et al., 2012; Chimuka et al., 2023; Jones et al., 2016; Koven et al., 2023; Melnikova et al., 2021; Schwinger and Tjiputra, 2018). During the period of decreasing atmospheric $CO_2$ concentrations and temperature (ramp-down), the β and γ feedbacks are influenced by both the reduction of $CO_2$ levels and temperature and the inertia of the carbon cycle—specifically, the altered land and ocean carbon pools resulting from prior increases in the $CO_2$ concentrations and temperature (Chimuka et al., 2023; Zickfeld et al., 2016).
* * *
**[Comment 8]**

   L69-71: This sentence is too long. For clarity, please separate the two research questions using (1) and (2) or a semi-colon.

**[Response]**

We followed the Reviewer's suggestion by moving "carbon cycle responses" to the first question and focusing on the nonlinearity feedback in the second research question.

The purpose of this study is twofold:

−   to clarify whether the climate and carbon cycle responses to declining $CO_2$ and non-

$CO_2$ GHGs differ globally and regionally

−   to investigate the carbon cycle nonlinearity feedback under $CO_2$ and non-$CO_2$ GHG

decrease, and the different implications for climate change mitigation.

**********

**[Comment 9]**

L81: Please clarify which climate factors you are referring to here.

**[Response]**

Following specific comment 5 of Reviewer #1, we removed this paragraph on the study's approach limitations, just keeping part of it in the discussion section. Thus, this sentence has now been deleted. We keep justification of the use of IPSL-CM6A-LR with the following text.

However for this study, the use of the model is justified because current changes in

$CH_4$ and $N_2O$ concentrations are primarily driven by anthropogenic sources, suggesting that the absence of interactive modules of natural sink/source processes does not significantly affect the representation of natural variability trends for the $CH_4$ and $N_2O$ concentration (Nakazawa, 2020; Palazzo Corner et al., 2023; Zhu et al., 2013).

**********

**[Comment 10]**

L120: From my understanding of the table format, experiments are above the horizontal line, while combinations of experiments are below the horizontal line. This is why I

am surprised that the [CO2bgc+non-CO2] experiment is above the line. Is this an experiment or an addition of two separately run experiments? If it is indeed an experiment, then I assume you prescribed both CO2 forcing and non-CO2 forcings, then specified the piControl CO2

concentration in the radiation code? If so, that would mean that the only warming seen in that experiment would be CO2 physiological warming, so how then can non-CO2 $\gamma$ be included in this experiment? Please clarify.

**[Response]**

This understanding is correct, this was indeed an experiment. We prescribed the piControl $CO_2$ concentration and varying non-$CO_2$ ($CH_4$ and $N_2O$) concentrations in the radiation code. Thus, the non-$CO_2$ radiative and $CO_2$ physiological (negligible) forcings caused the warming. This is consistent with our original description in the table. We added a clarifying sentence to the section on Experiment design.

Additionally, an experiment that combines non$CO_2$ radiative forcing with $CO_2$ physiological forcing [$CO_2$bgc + non$CO_2$] allows for the comparison of nonlinearities arising from combined carbon-concentration feedback and $CO_2$- and non-$CO_2$-driven carbon-climate feedback ([$CO_2$bgc + non$CO_2$]). It serves as the non$CO_2$ counterpart of the [$CO_2$] experiment.
* * *
**[Comment 11]**

On the same note, is the additional combination [CO2bgc+non-CO2]-[CO2bgc] necessary? It looks like we could get at non-CO2 $\gamma$ by taking the difference between [CO2+non-CO2] and [CO2] and this would give the cross term as well. Is there a benefit to using [CO2bgc+non-CO2]-[CO2bgc] over [CO2+non-CO2]-[CO2]?

**[Response]**

The Reviewer is correct that non-CO2 $\gamma$ may be derived either from [CO2bgc+non-CO2]-[CO2bgc] or from [CO2+non-CO2]-[CO2], with both combinations involving two experiments. However, there are at least two benefits of using [CO2bgc+non-CO2]-[CO2bgc]. Firstly, it is consistent with deriving $\gamma_{CO2}$ and $\chi_{CO2}$ terms from [CO2]-[CO2bgc], because both combinations subtract the BGC component from an experiment that has $\beta$, $\gamma$ and $\chi$. Secondly, using [CO2+non-CO2]-[CO2] would lead to a using an experiment with nearly doubled warming level ([CO2+non-CO2]), that would affect the value of cross term $\chi$ (probably by overestimating it).
* * *
**[Comment 12]**

In the 4th column, the first two combinations of experiments seem to be missing the $\Delta U$! components.

**[Response]**

Thank you, this is now corrected.

**********

**[Comment 13]**

Figure 1: I would like to commend the authors on this figure – it complements the
methods section very nicely.
**[Response]**

We sincerely thank the Reviewer.

**********

**[Comment 14]**

L146: Section 3.1 assumes that readers have a solid grasp of the carbon cycle
feedback framework and the feedback parameters $(\beta, \gamma)$ used, which may not be the case. I
suggest prefacing this section with a brief description of carbon cycle feedback parameters
(equations for quantification, units and sign convention) before introducing $\Delta U$.
**[Response]**

We added a paragraph with a brief explanation on the $\beta, \gamma$ quantification, units and
sign convention, as suggested (although not before but after introducing $\Delta U$), as described in
our response to the Reviewer's main comment.

**********

**[Comment 15]**

L184: I suggest citing Zickfeld et al. (2011) here.
**[Response]**

Thank you for bringing up this study that we had missed. The refence has now be
added together with some other relevant publications that we also missed in the original
manuscript (Arora et al., 2013; Schwinger et al., 2014). We also added references to Zickfeld
et al. (2011) in several other places in the revised manuscript (e.g., in the new paragraph in
the Introduction about existing studies on nonlinearity of carbon cycle feedbacks).

\*\*\*\*\*\*\*\*\*\*

**[Comment 16]**

Figure 2: Is the last column of panels on Figure 2 necessary? I notice that these figures
are hardly referenced.

**[Response]**

We agree and removed the last column of Figure 2.

\*\*\*\*\*\*\*\*\*\*

**[Comment 17]**

Also, I suggest using a different colour for either the CO2 or CO2bgc lines? The two
are compared several times in the text but the colours are difficult to distinguish on the figure
panels.

**[Response]**

We changed the colour of [$CO_2$] from orange to deep pink for a better distinction.

\*\*\*\*\*\*\*\*\*\*

**[Comment 18]**

L219: What is the reason for the higher sensitivity to non-CO2 forcing than CO2
forcing?

**[Response]**

We apologize for the confusion in text, as it should be opposite, i.e., higher sensitivity
of $CO_2$ forcing compared to non-$CO_2$ forcing. We made the correction and added a clarification
for the reason as follows.

Our results are consistent with Nordling et al. (2021) who show the higher effective
temperature response for $CO_2$ forcing compared to non-$CO_2$ forcing, attributing it to the
changes in clear-sky planetary emissivity.

\*\*\*\*\*\*\*\*\*\*

**[Comment 19]**

L262: It appears that the figure in the paper referenced – Chimuka et al. (2023) – shows little hysteresis in autotrophic respiration and GPP, and not in heterotrophic respiration as mentioned in the text.

**[Response]**

This is indeed true, we misread the paper. We now removed the sentence.

\*\*\*\*\*\*\*\*\*\*

**[Comment 20]**

L283-284: Are there merits to attributing the cross term to $\gamma$ rather than keeping it as a separate term?

Keeping the legacy of previous studies is probably the biggest merit. However, considering the implications of the carbon cycle framework for nonCO$_2$ scenarios, it is more accurate to keep the cross-term as a separate feedback term. Following encouragement from Reviewer #1, we introduced the new symbol χ for the cross-term and divided the original "Carbon-Climate Feedback" section into two parts, creating a new section titled "Nonlinearity in Carbon Cycle Feedback."

**References**

Arora, V. K., Boer, G. J., Friedlingstein, P., Eby, M., Jones, C. D., Christian, J. R., Bonan, G.,
Bopp, L., Brovkin, V., Cadule, P., Hajima, T., Ilyina, T., Lindsay, K., Tjiputra, J. F., and Wu, T.:
Carbon–Concentration and Carbon–Climate Feedbacks in CMIP5 Earth System Models, J.
Climate, 26, 5289–5314, https://doi.org/10.1175/JCLI-D-12-00494.1, 2013.

Arora, V. K., Katavouta, A., Williams, R. G., Jones, C. D., Brovkin, V., Friedlingstein, P.,
Schwinger, J., Bopp, L., Boucher, O., Cadule, P., Chamberlain, M. A., Christian, J. R., Delire,
C., Fisher, R. A., Hajima, T., Ilyina, T., Joetzjer, E., Kawamiya, M., Koven, C. D., Krasting, J.
P., Law, R. M., Lawrence, D. M., Lenton, A., Lindsay, K., Pongratz, J., Raddatz, T., Séférian,
R., Tachiiri, K., Tjiputra, J. F., Wiltshire, A., Wu, T., and Ziehn, T.: Carbon–concentration and
carbon–climate feedbacks in CMIP6 models and their comparison to CMIP5 models,
Biogeosciences, 17, 4173–4222, https://doi.org/10.5194/bg-17-4173-2020, 2020.

Asaadi, A., Schwinger, J., Lee, H., Tjiputra, J., Arora, V., Séférian, R., Liddicoat, S., Hajima,
T., Santana-Falcón, Y., and Jones, C. D.: Carbon cycle feedbacks in an idealized simulation
and a scenario simulation of negative emissions in CMIP6 Earth system models,
Biogeosciences, 21, 411–435, https://doi.org/10.5194/bg-21-411-2024, 2024.

Boucher, O., Halloran, P. R., Burke, E. J., Doutriaux-Boucher, M., Jones, C. D., Lowe, J.,
Ringer, M. A., Robertson, E., and Wu, P.: Reversibility in an Earth System model in response
to CO 2 concentration changes, Environ. Res. Lett., 7, 024013, https://doi.org/10.1088/1748-
9326/7/2/024013, 2012.

Chimuka, V. R., Nzotungicimpaye, C.-M., and Zickfeld, K.: Quantifying land carbon cycle
feedbacks under negative $CO_2$ emissions, Biogeosciences, 20, 2283–2299,
https://doi.org/10.5194/bg-20-2283-2023, 2023.

Friedlingstein, P., Dufresne, J.-L., Cox, P. M., and Rayner, P.: How positive is the feedback
between climate change and the carbon cycle?, Tellus B: Chemical and Physical Meteorology,
https://doi.org/10.3402/tellusb.v55i2.16765, 2003.

Friedlingstein, P., Cox, P., Betts, R., Bopp, L., von Bloh, W., Brovkin, V., Cadule, P., Doney,
S., Eby, M., Fung, I., Bala, G., John, J., Jones, C., Joos, F., Kato, T., Kawamiya, M., Knorr,
W., Lindsay, K., Matthews, H. D., Raddatz, T., Rayner, P., Reick, C., Roeckner, E., Schnitzler,
K.-G., Schnur, R., Strassmann, K., Weaver, A. J., Yoshikawa, C., and Zeng, N.: Climate–
Carbon Cycle Feedback Analysis: Results from the C4MIP Model Intercomparison, J. Climate,
19, 3337–3353, https://doi.org/10.1175/JCLI3800.1, 2006.

Gregory, J. M., Jones, C. D., Cadule, P., and Friedlingstein, P.: Quantifying Carbon Cycle
Feedbacks, J. Climate, 22, 5232–5250, https://doi.org/10.1175/2009JCLI2949.1, 2009.

Jones, C. D., Ciais, P., Davis, S. J., Friedlingstein, P., Gasser, T., Peters, G. P., Rogelj, J.,
van Vuuren, D. P., Canadell, J. G., Cowie, A., Jackson, R. B., Jonas, M., Kriegler, E., Littleton,
E., Lowe, J. A., Milne, J., Shrestha, G., Smith, P., Torvanger, A., and Wiltshire, A.: Simulating
the Earth system response to negative emissions, Environmental Research Letters, 11,
095012, https://doi.org/10.1088/1748-9326/11/9/095012, 2016.

Koven, C. D., Sanderson, B. M., and Swann, A. L. S.: Much of zero emissions commitment
occurs before reaching net zero emissions, Environmental Research Letters, 18, 014017,
https://doi.org/10.1088/1748-9326/acab1a, 2023.

Melnikova, I., Boucher, O., Cadule, P., Ciais, P., Gasser, T., Quilcaille, Y., Shiogama, H.,
Tachiiri, K., Yokohata, T., and Tanaka, K.: Carbon cycle response to temperature overshoot beyond 2 °C – an analysis of CMIP6 models, Earth's Future, 9, e2020EF001967,
https://doi.org/10.1029/2020EF001967, 2021.

Nakazawa, T.: Current understanding of the global cycling of carbon dioxide, methane, and
nitrous oxide, Proceedings of the Japan Academy, Series B, 96, 394–419,
https://doi.org/10.2183/pjab.96.030, 2020.

Nordling, K., Korhonen, H., Räisänen, J., Partanen, A.-I., Samset, B. H., and Merikanto, J.:
Understanding the surface temperature response and its uncertainty to CO2, CH4, black
carbon, and sulfate, Atmos. Chem. Phys., 21, 14941–14958, https://doi.org/10.5194/acp-21-
14941-2021, 2021.

Palazzo Corner, S., Siegert, M., Ceppi, P., Fox-Kemper, B., Frölicher, T. L., Gallego-Sala, A.,
Haigh, J., Hegerl, G. C., Jones, C. D., Knutti, R., Koven, C. D., MacDougall, A. H.,
Meinshausen, M., Nicholls, Z., Sallée, J. B., Sanderson, B. M., Séférian, R., Turetsky, M.,
Williams, R. G., Zaehle, S., and Rogelj, J.: The Zero Emissions Commitment and climate
stabilization, Frontiers in Science, 1, https://doi.org/10.3389/fsci.2023.1170744, 2023.

Richardson, T. B., Forster, P. M., Smith, C. J., Maycock, A. C., Wood, T., Andrews, T., Boucher,
O., Faluvegi, G., Fläschner, D., Hodnebrog, Ø., Kasoar, M., Kirkevåg, A., Lamarque, J.-F.,
Mülmenstädt, J., Myhre, G., Olivié, D., Portmann, R. W., Samset, B. H., Shawki, D., Shindell,
D., Stier, P., Takemura, T., Voulgarakis, A., and Watson-Parris, D.: Efficacy of Climate
Forcings in PDRMIP Models, J Geophys Res Atmos, 124, 12824–12844,
https://doi.org/10.1029/2019JD030581, 2019.

Schwinger, J. and Tjiputra, J.: Ocean Carbon Cycle Feedbacks Under Negative Emissions,
Geophysical Research Letters, 45, 5062–5070, https://doi.org/10.1029/2018GL077790, 2018.

Schwinger, J., Tjiputra, J. F., Heinze, C., Bopp, L., Christian, J. R., Gehlen, M., Ilyina, T., Jones,
C. D., Salas-Mélia, D., Segschneider, J., Séférian, R., and Totterdell, I.: Nonlinearity of Ocean
Carbon Cycle Feedbacks in CMIP5 Earth System Models, Journal of Climate, 27, 3869–3888,
https://doi.org/10.1175/JCLI-D-13-00452.1, 2014.

Williams, R. G., Katavouta, A., and Goodwin, P.: Carbon-Cycle Feedbacks Operating in the
Climate System, Current Climate Change Reports, 5, 282–295,
https://doi.org/10.1007/s40641-019-00144-9, 2019.

Zhu, X., Zhuang, Q., Gao, X., Sokolov, A., and Schlosser, C. A.: Pan-Arctic land–atmospheric
fluxes of methane and carbon dioxide in response to climate change over the 21st century,
Environmental Research Letters, 8, 045003, https://doi.org/10.1088/1748-9326/8/4/045003,
2013.

Zickfeld, K., Eby, M., Matthews, H. D., Schmittner, A., and Weaver, A. J.: Nonlinearity of
Carbon Cycle Feedbacks, Journal of Climate, 24, 4255–4275,
https://doi.org/10.1175/2011JCLI3898.1, 2011.

Zickfeld, K., MacDougall, A. H., and Matthews, H. D.: On the proportionality between global
temperature change and cumulative CO 2 emissions during periods of net negative CO 2
emissions, Environ. Res. Lett., 11, 055006, https://doi.org/10.1088/1748-9326/11/5/055006,
2016.

---

## Editor Decision (ED1)

While the latest revisions in response to reviewer comments have helped to clarify the manuscript, further clarifications in some instances would be helpful (see specific comments attached). I recommend that the authors re-read the manuscript carefully with particular attention to clarity to make it as easy as possible to the reader to follow.

l. 41: A reference to this recent study is missing:
Nzotungicimpaye et al., 2023, Delaying methane mitigation increases the risk of breaching the 2°C warming limit. Communications Earth and Environment 4, 250, https://doi.org/10.1038/s43247-023-00898-z

l. 58-60: Nzotungicimpaye et al., 2023 also discusses the effect of methane mitigation on the carbon cycle.

l. 69 "nonlinearity in the system": cite Schwinger et al., 2014; Zickfeld et al., 2011 again here.

l. 157 "Three ensemble members…": Clarify whether three ensemble members are run for each experiment.

l. 175-176: "The beta feedback reflects the strengthening…": This is only true for rising atmospheric CO2 concentrations. I suggest to word this in a neutral way that applies to both increasing and decreasing atmospheric CO2.

l. 176 "positive response": Avoid value judgements in this context as there is a risk of confusion with the sign of the feedback.

l.180-181: "The gamma feedback reflects the weakening…": This is only true for rising temperature. As for beta, I suggest to word this in a neutral way that applies to both warming and cooling.

l. 267-268: This sentence is confusing. Why not say "Radiative forcing alone ([CO2rad] experiment) leads to a slightly higher global temperature increase compared to the coupled [CO2] experiment, which includes the combined effect of CO2 physiology and radiative forcing".

l. 314: Fig. 3 caption: Point out that vertical axes differ between panels e, g and i, j.

l. 322: "… ocean carbon sink". I think this should read "ocean carbon source".

l.323-324: "It is nearly equivalent …". I wonder if the difference between land and ocean is merely due to the different vertical scales used in Fig. 3 panels e, g?

l.324: Should refer to panels e, g (not f, g).

l. 334-335 "greater reduction in the climate-driven carbon sink": In my mind this should read "greater reduction in the $CO_2$-driven sink". Climate (warming) drives a source, whereas rising atmospheric $CO_2$ drives a sink.

l. 350-352. Clarify which experiment you are referring to. I suppose it's [CO2-BGC] and [non-CO2]?

Edits/typos:
l. 64 and elsewhere: "over the ocean" should read "in the ocean".
l. 100: "runup for" → "runup to"
l. 278: delete extra "in".
l. 335: delete extra "driven".
l. 413: "priority to" → priority over".
l. 413: insert "they" before "provide".

---

## Author Response (AR2)

Our responses are in black, marked as **[Response]**, and the comments of the
Reviewers are in purple, marked as **[Comment]**. In our responses, we mark the
changes in the manuscript with shading and separate comments using "***********".

**Reviewer #1 (Remarks to the Author):**

The authors did a good job in revising their manuscript, and I think it can be accepted
for publication after a few minor things (listed below) have been addressed.

We thank Dr. Schwinger for taking the time to read the manuscript once again
and providing positive feedback on the revised manuscript with additional comments
to improve the manuscript.

**********

**General comments**

**[Comment 1]**

Section 2.3: I'm not entirely convinced about how this section is structured, and
the wording is sometimes a bit misleading.
The carbon uptake is not "estimated" (line 278), it is derived from the
simulations. What is "estimated" are the feedback factors. Equations 3 and 4 are very
confusing since equation 4 is not gamma but gamma+chi according to the framework
introduced later. Why not begin this section with the Taylor expansion? This is the
basis of the whole framework, and it would make it clear that gamma actually is defined
at constant CO2 (i.e. we need the RAD simulation to determine it) gamma=(del U / del
T)_{CO2=const}. The definition of the cross term would also become clear from the
beginning, and that the COU-BGC simulation includes the cross term.
Regarding Eq. 7,8,9 I still would favor to omit the quadratic terms, or it should
at least be made clearer that, when estimating the feedback factors from simulations,
the estimate includes the quadratic terms. You say "we found them [the residual terms]
to be negligible" How is it possible to determine them (you can't even determine the
quadratic terms, if I'm not mistaken).

**[Response]**

We originally added the preface in this section, following a comment from Reviewer 2 who argued that the Section "assumes that readers have a solid grasp of the carbon cycle feedback framework and the feedback parameters ($\beta$, $\gamma$) used, which may not be the case." Reviewer 2 suggested to add a brief description of carbon cycle feedback parameters, equations for quantification, units and sign convention before introducing the Taylor expansion. We tend to agree with this argument and thus keep the preface in a shortened form. Particularly, we removed Eq. (4) ($\gamma = \frac{\Delta U_{COU-BGC}}{\Delta T}$) that may be confusing in regard to the framework introduced later and a paragraph describing the experiments with the corresponding terms.

We follow your suggestion regarding the second-order terms in Eq. (6) and (7) (with revised numbering). The text was changed to the following:

$$\Delta U_\beta = \frac{\partial U}{\partial C_{CO2}} \Delta C_{CO2} + Res., \tag{6}$$

$$\Delta U_\gamma = \frac{\partial U}{\partial T} \Delta T + Res., \tag{7}$$

…

For simplicity, the second-order terms of Eqs. (6) and (7) are included in $Res.$.

**[Comment 2]**

Many places in the manuscript, the authors use an imprecise wording regarding the feedback factors. For example (line 316), "During the ramp-up gamma drives a carbon sink…". It is climate warming that drives the carbon sink, and a positive value of gamma is the consequence. Beta, gamma and chi are diagnostic quantities, they are not drivers. I would encourage the authors to go once more through the results section and reword this and similar sentences (e.g., but not limited to, lines 327, 334ff, 339, 344).

**[Response]**

We revised our wording throughout the manuscript, especially in Sections 3.3 – 3.4 that show results on the carbon-concentration, carbon-climate and nonlinearity in carbon cycle feedback.

**Specific/technical comments**

**[Comment 1]**

line 20: "influence only the carbon-climate feedback" would read better "gives rise to a carbon climate feedback only"

**[Response]**

Changed as suggested.
* * *
**[Comment 2]**

line 21: "… however, focused exclusively on CO2 forcing" sounds strange, please consider rewording (or deleting since it seems not really necessary in an abstract)

**[Response]**

We reworded to:

We introduce a framework, building on previous studies that primarily addressed $CO_2$ forcing, to separate the carbon-climate feedback into a temperature term and a temperature–$CO_2$ cross term.
* * *
**[Comments 3–6]**

line 22: the term "cross term" is not self-explanatory. Maybe better say "… into a temperature and a temperature-CO2 cross term" or similar.

line 35: of → over line 62: "are associated" → maybe better "give rise to a carbon-concentration feedback" (I believe using the symbol beta without further introducing it here is not necessary).

line 65: same comment as for line 62

Changed as suggested.

**********

**[Comment 7]**

line 75: Since "the cross term" is introduced here it would be good to say what this means, e.g., "the cross term arising from interactions of changing atmospheric

CO2 and changing temperatures"

**[Response]**

Changed, now reads:

Previous studies investigated the nonlinearity in the carbon cycle feedback, showing that the cross term—arising from interactions between changing atmospheric

$CO_2$ and temperatures—can be comparable in size with $\gamma$.

**********

**[Comment 8]**

line 93: consider changing to "… to investigate the nonlinearities of carbon cycle feedbacks …"; consider deleting "different".

**[Response]**

Changed as suggested.

**********

**[Comment 9]**

line 130: "that includes only CO2 physiological forcing" this is confusing for the reader. First and foremost this experiment includes CO2 forcing that is only seen by the land and ocean. Then there is also a small temperature forcing due to the CO2

physiological forcing. Please consider rewording.

**[Response]**

Changed to:

…a biogeochemically coupled (BGC) experiment where $CO_2$ forcing affects only the carbon cycle of land and ocean [$CO_2$bgc] (with minor temperature effects from $CO_2$ physiological forcing)…
* * *
**[Comment 10–12]**

line 133: "feedback nonlinearities" → "nonlinearities of feedbacks"

line 144: it is the ERF that is estimated from Etminan et al. not the concentrations, right? If so please consider moving this after "3.69 W m-2" other wise it is confusing.

line 153: "a-c panels" → "panels a-c"

Changed as suggested.
* * *
**[Comment 13]**

line 221: maybe worth noting that Asaadi et al. 2024 found that the effect of the warming on beta is indeed negligible

**[Response]**

We added the citation.

…consistent with findings of Asaadi et al., (2024).
* * *
**[Comments 14–15]**

line 242: "maximum" → "strongest decrease"

line 344: "compared to when atmospheric CO2 is constant" → "compared to the RAD experiment, in which atmospheric CO2 is constant"

**[Response]**

Changed as suggested.
* * *
**[Comment 16]**

line 354: Should "carbon-concentration" read "carbon-climate"?

**[Response]**

We removed the clarification in brackets (originally "which involve carbon-
concentration feedback alterations") altogether, as it is thoroughly explained in the
following sentence.

**********

**[Comments 17–20]**

line 355: delete "concentration"? I find it confusing in this context.

Figure 4: I would suggest to change the y-axis labels from "Delta U_{beta
gamma}" to Delta U_{Chi}

line 410: "the presence of carbon concentration feedback…" would be better
worded as "increasing atmospheric CO2 amplifies the reduction of the climate change
driven line 412: maybe better: "… and a component driven by climate change and
rising atmospheric CO2 at the same time, i.e. a cross term."

**[Response]**

Changed as suggested.

**********

**Reviewer #2 (Remarks to the Author):**

The manuscript is in good shape. I would just suggest one change for clarity: changing the wording in the second research question from "carbon cycle non-linearity feedback" to "carbon cycle non-linearity" (Lines 101-103 in the tracked changes version of the manuscript).

**[Response]**

We thank the Reviewer for the positive feedback of the revised manuscript. We made the change for clarity, and now it reads "the nonlinearities of carbon cycle feedbacks".

---

## Author Response (AR3)

Our responses are in black, marked as **[Response]**, and the comments of the Editor are in purple, marked as **[Comment]**. In our responses, we mark the changes in the manuscript with shading and separate comments using "**********".

**Editor (Remarks to the Author):**

While the latest revisions in response to reviewer comments have helped to clarify the manuscript, further clarifications in some instances would be helpful (see specific comments attached). I recommend that the authors re-read the manuscript carefully with particular attention to clarity to make it as easy as possible to the reader to follow.

We thank Dr. Zickfeld for taking the time to read the manuscript and providing additional comments to improve it. We have followed the recommendation, gone through the manuscript and made small changes to make it easier for the reader to follow. We hope that the revised manuscript is now ready for publication in *ESD*.
* * *
**[Comment 1]**

l. 41: A reference to this recent study is missing:

Nzotungicimpaye et al., 2023, Delaying methane mi8ga8on increases the risk of breaching the 2°C warming limit. Communica8ons Earth and Environment 4, 250, h;ps://doi.org/10.1038/s43247-023-00898-z

**[Response]**

Thank you for pointing out the study that we missed. We added the reference.

**[Comment 2]**

l. 58-60: Nzotungicimpaye et al., 2023 also discusses the effect of methane mitigation on the carbon cycle

**[Response]**

We added the summary of research findings of Nzotungicimpaye et al., 2023 to the introduction as follows:

Using intermediate-complexity Earth System Climate Model simulations, Nzotungicimpaye et al. (2023) showed that delaying methane mitigation has implications both for meeting the stringent temperature targets and for the climate over many centuries.

**[Comment 3]**

l. 69 "nonlinearity in the system": cite Schwinger et al., 2014; Zickfeld et al., 2011 again here.

**[Response]**

Citation is now added.

**[Comment 4]**

l. 157 "Three ensemble members…": Clarify whether three ensemble members are run for each experiment.

**[Response]**

Clarification added: "For each experiment, three ensemble members". We have also added the following clarifying text:

We note that the use of three members is not ideal, but it is a common compromise between computational cost and sampling the uncertainty due to climate variability.

**[Comment 5]**

l. 175-176: "The beta feedback reflects the strengthening…": This is only true for rising atmospheric CO2 concentrations. I suggest to word this in a neutral way that applies to both increasing and decreasing atmospheric CO2.

**[Response]**

We agree and simplified the text to:

The β feedback reflects the changes in land and ocean carbon pools driven by the changes in $CO_2$ concentrations.

**[Comment 6]**

l. 176 "positive response": Avoid value judgements in this context as there is a risk of confusion with the sign of the feedback.

**[Response]**

We removed the sentence from the revised manuscript.

**[Comment 7]**

l.180-181: "The gamma feedback reflects the weakening…": This is only true for rising temperature. As for beta, I suggest to word this in a neutral way that applies to both warming and cooling.

**[Response]**

We agree and simplified the text to:

The γ feedback reflects the changes in the land and ocean carbon pools due to the changes in climate.

**[Comment 8]**

l. 267-268: This sentence is confusing. Why not say "Radiative forcing alone ([CO2rad] experiment) leads to a slightly higher global temperature increase compared to the coupled [CO2] experiment, which includes the combined effect of CO2 physiology and radiative forcing".

**[Response]**

We think the Editor refers to lines 257-258. We agree that the suggested change
makes the statement clearer. Changed accordingly.

**[Comment 9]**
l. 314: Fig. 3 caption: Point out that vertical axes differ between panels e, g and i, j.
**[Response]**

We added clarification to the caption: Note that vertical axes differ between panels e,
g and i, j.

**[Comment 10]**
l. 322: "… ocean carbon sink". I think this should read "ocean carbon source".

**[Response]**

Corrected.

**[Comment 11]**
l.323-324: "It is nearly equivalent …". I wonder if the difference between land and
ocean is merely due to the different vertical scales used in Fig. 3 panels e, g?

**[Response]**

We understand the Editor's concern. With only a three-member ensemble, it is
challenging to confirm differences statistically. The quantified differences for χ, as
shown in Tables 2 and S1, suggest a larger difference for the ocean, partly due to the
higher uncertainty associated with the land. We have revised the text as follows for a
more careful statement to avoid misinterpretation:

The χ feedback is positive (larger carbon sink) in the land and negative (larger carbon source) in the ocean (Table S1). There is no significant difference between $CO_2$ and non-$CO_2$ χ feedback at similar ERF levels (Fig. 3e-j, Fig. 4 f-g, Tables 2 and S1).

**[Comment 12]**

l.324: Should refer to panels e, g (not f, g).

**[Response]**

Originally, we refer to panels f, g of Fig. 4 (showing spatial variation). We guess that the Editor refer to panels e, g of Fig. 3. We agree that panels e, g (also i and j) are relevant to our statement. We changed text to have all: (Fig. 3e-j, Fig. 4 f-g, Tables 2 and S1).

**[Comment 13]**

l. 334-335 "greater reduction in the climate-driven carbon sink": In my mind this should read "greater reduction in the CO2-driven sink". Climate (warming) drives a source, whereas rising atmospheric CO2 drives a sink.

**[Response]**

We agree, changed accordingly.

**[Comment 14]**

l. 350-352. Clarify which experiment you are referring to. I suppose it's [CO2-BGC] and [non-CO2]?

**[Response]**

We refer to $[CO_2] – [CO_2bgc]$ and $[nonCO_2]$, clarified in the revised manuscript.

… (compare red and black lines in Fig. 3, corresponding to [$CO_2$] – [$CO_2bgc$] and

[$nonCO_2$] experiments) …

**Edits/typos:**

l. 64 and elsewhere: "over the ocean" should read "in the ocean".

l. 100: "runup for" à "runup to"

l. 278: delete extra "in".

l. 335: delete extra "driven".

l. 413: "priority to" à priority over".

l. 413: insert "they" before "provide".

**[Response]**

Corrected.